# Stronger NAS with Weaker Predictors

**Junru Wu[1], Xiyang Dai[2], Dongdong Chen[2], Yinpeng Chen[2], Mengchen Liu[2], Ye Yu[2],
Zhangyang Wang[3], Zicheng Liu[2], Mei Chen[2], Lu Yuan[2]**
[1] Texas A&M University, [2]Microsoft Corporation, [3]University of Texas at Austin
sandboxmaster@tamu.edu, {xidai,dochen,yilche,mengcliu}@microsoft.com,
atlaswang@utexas.edu, {yeyu1,zliu,mei.chen,luyuan}@microsoft.com

## Abstract

Neural Architecture Search (NAS) often trains and evaluates a large number of architectures. Recent predictor-based NAS approaches attempt to alleviate such heavy computation costs with two key steps: sampling some architecture-performance pairs and fitting a proxy accuracy predictor. Given limited samples, these predictors, however, are far from accurate to locate top architectures due to the difficulty of fitting the huge search space. This paper reflects on a simple yet crucial question: *if our final goal is to find the best architecture, do we really need to model the whole space well?*. We propose a paradigm shift from fitting the whole architecture space using one strong predictor, to progressively fitting a search path towards the high-performance sub-space through a set of weaker predictors. As a key property of the weak predictors, their probabilities of sampling better architectures keep increasing. Hence we only sample a few well-performed architectures guided by the previously learned predictor and estimate a new better weak predictor. This embarrassingly easy framework, dubbed **WeakNAS**, produces coarse-to-fine iteration to gradually refine the ranking of sampling space. Extensive experiments demonstrate that WeakNAS costs fewer samples to find top-performance architectures on NAS-Bench-101 and NAS-Bench-201. Compared to state-of-the-art (SOTA) predictor-based NAS methods, WeakNAS outperforms all with notable margins, e.g., requiring **at least 7.5x** less samples to find global optimal on NAS-Bench-101. WeakNAS can also absorb their ideas to boost performance more. Further, Weak-NAS strikes the new SOTA result of 81.3% in the ImageNet MobileNet Search Space. The code is available at: `https://github.com/VITA-Group/WeakNAS`.

## 1 Introduction

Neural Architecture Search (NAS) [1–12] methods aim to find the best network architecture by exploring the architecture-to-performance manifold, using reinforced-learning-based [13], evolution-based [14, 15] or gradient-based [1, 16] approaches. In order to cover the entire search space, they often train and evaluate a large number of architectures, leading to tremendous computation cost.

Recently, predictor-based NAS methods alleviate this problem with two key steps: one sampling step to sample some architecture-performance pairs, and another performance modeling step to fit the performance distribution by training a proxy accuracy predictor. An in-depth analysis of existing methods [2] found that most of those methods [5, 6, 17, 7–9, 18] consider these two steps independently and attempt to model the performance distribution over the whole architec-

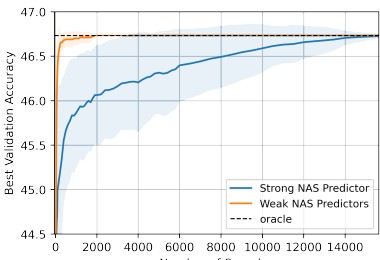

Figure 1: Comparison between our method using a set of weak predictors (iterative sampling), and a single strong predictor (random sampling) on NAS-Bench-201. For fair comparison, the NAS predictor in both methods adtops the same type of MLP described in 2.4. Solid lines and shadows denote the mean and standard deviation (std), respectively.

35th Conference on Neural Information Processing Systems (NeurIPS 2021).

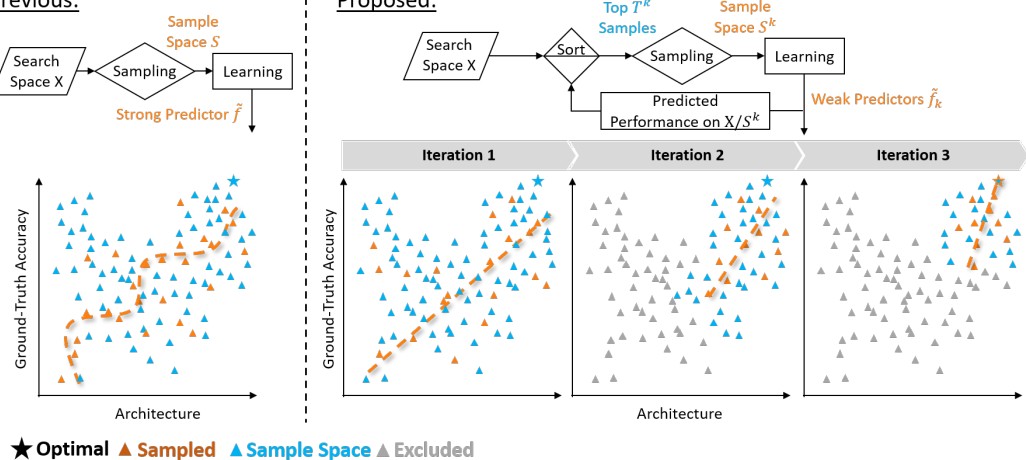

Figure 2: An illustration of WeakNAS's progressive approximation. Previous predictor-based NAS uniformly sample in the whole search space to fit a strong predictor. Instead, our method progressively shrinks the sample space based on predictions from previous weak predictors, and update new weak predictors towards subspace of better architectures, hence focusing on fitting the search path.

ture space using a ***strong***[1] predictor. However, since the architecture space is often exponentially large and highly non-convex, even a very strong predictor model has difficulty fitting the whole space given limited samples. Meanwhile, different types of predictors often demand handcraft design of the architecture representations to improve their performance.

This paper reflects on a fundamental question for predictor-based NAS: *"if our final goal is to find the best architecture, do we really need to model the whole space well?"*. We investigate the alternative of utilizing a few ***weak***[1] predictors to fit small local spaces, and to progressively move the search space towards the subspace where good architecture resides. Intuitively, we assume the whole space could be divided into different sub-spaces, some of which are relatively good while some are relatively bad. We tend to choose the good ones while discarding the bad ones, which makes sure more samples will be focused on modeling only the good subspaces and then find the best architecture. It greatly simplifies the learning task of each predictor. Eventually, a line of progressively evolving weak predictors can connect a path to the best architecture.

We present a novel, general framework that requires only to estimate a series of weak predictors progressively along the search path, we denoted it as **WeakNAS** in the rest of the paper. To ensure moving towards the best architecture along the path, at each iteration, the sampling probability of better architectures keep increasing through the guidance of the previous weak predictor. Then, the consecutive weak predictors with better samples will be trained in the next iteration. We iterate until we arrive at an embedding subspace where the best architectures reside and can be accurately assessed by the final weak predictor.

Compared to the existing predictor-based NAS, our proposal represents a new line of attack and has several merits. First, since only weak predictors are required, it yields better sample efficiency. As shown in Figure 1, it costs significantly fewer samples to find the top-performance architecture than using one strong predictor, and yields much lower variance in performance over multiple runs. Second, it is flexible to the choices of architecture representation (e.g., different architecture embeddings) and predictor formulation (e.g., multilayer perceptron (MLP), gradient boosting regression tree, or random forest). Experiments show our framework performs well in all their combinations. Third, it is highly generalizable to other open search spaces, e.g. given a limited sample budget, we achieve the state-of-the-art ImageNet performance on the NASNet and MobileNet search spaces. Detailed comparison with state-of-the-art predictor-based NAS [19–21, 8] is presented in Section 4.

---

[1]"Strong" vs "Weak" predictors: we name a "weak" predictor if it only predicts a local subspace of the search space thus can be associated with our iterative sampling scheme; such predictors therefore usually do not demand very heavily parameterized models. On the contrary, "strong" predictors predict the global search space and are often associated with uniform sampling. The terminology of strong versus weak predictors does not represent their number of parameters or the type of NAS predictor used. An overparameterized NAS predictor with our iterative sampling scheme may still be considered as a "weak" predictor.

## 2 Our Framework

### 2.1 Reformulating Predictor-based NAS as Bi-Level Optimization

Given a search space of network architectures $X$ and an architecture-to-performance mapping function $f : X \to P$ from the architecture set $X$ to the performance set $P$, the objective is to find the best neural architecture $x^*$ with the highest performance $f(x)$ in the search space $X$:

$$x^* = \arg\max_{x \in X} f(x) \tag{1}$$

A naïve solution is to estimate the performance mapping $f(x)$ through the full search space. However, this is prohibitively expensive since all architectures have to be exhaustively trained from scratch. To address this problem, predictor-based NAS learns a proxy predictor $\tilde{f}(x)$ to approximate $f(x)$ by using some architecture-performance pairs, which significantly reduces the training cost. In general, predictor-based NAS can be re-cast as a bi-level optimization problem:

$$x^* = \arg\max_{x \in X} \tilde{f}(x|S), \ \text{s.t.} \ \tilde{f} = \arg\min_{S, \tilde{f} \in \tilde{\mathcal{F}}} \sum_{s \in S} \mathcal{L}(\tilde{f}(s), f(s)) \tag{2}$$

where $\mathcal{L}$ is the loss function for the predictor $\tilde{f}$, $\tilde{\mathcal{F}}$ is a set of all possible approximation to $f$, $S := \{S \subseteq X \mid |S| \le C\}$ all architectures satisfying the sampling budget $C$. $C$ is directly related to the total training cost, e.g., the total number of queries. Our objective is to minimize the loss $\mathcal{L}$ based on some sampled architectures $S$.

Previous predictor-based NAS methods attempt to solve Equation 2 with two sequential steps: (1) *sampling* some architecture-performance pairs and (2) *learning* a proxy accuracy predictor. For the first step, a common practice is to sample training pairs $S$ uniformly from the search space $X$ to fit the predictor. Such sampling is however inefficient considering that the goal of NAS is only to find well-performed architectures without caring for the bad ones.

### 2.2 Progressive Weak Predictors Emerge Naturally as A Solution to the Optimization

**Optimization Insight:** Instead of first (uniformly) sampling the whole space and then fitting the predictor, we propose to jointly evolve the sampling $S$ and fit the predictor $\tilde{f}$, which helps achieve better sample efficiency by focusing on only relevant sample subspaces. That could be mathematically formulated as solving Equation 2 in a new coordinate descent way, that iterates between optimizing the architecture *sampling* and predictor *fitting* subproblems:

$$\text{(Sampling)} \quad \tilde{P}^k = \{\tilde{f}_k(s)|s \in X \setminus S^k\}, \ S_M \subset \text{Top}_N(\tilde{P}^k), \ S^{k+1} = S_M \cup S^k, \tag{3}$$
$$\text{where } \text{Top}_N(\tilde{P}^k) \text{ denote the set of top N architectures in } \tilde{P}^k$$

$$\text{(Predictor Fitting)} \quad x^* = \arg\max_{x \in X} \tilde{f}(x|S^{k+1}), \ \text{s.t.} \ \tilde{f}_{k+1} = \arg\min_{\tilde{f}_k \in \tilde{\mathcal{F}}} \sum_{s \in S^{k+1}} \mathcal{L}(\tilde{f}(s), f(s)) \tag{4}$$

In comparison, existing predictor-based NAS methods could be viewed as running the above coordinate descent *for just one iteration* – a special case of our general framework.

As well known in optimization, many iterative algorithms only need to solve (subsets of) their subproblems inexactly [22–24] for properly ensuring convergence either theoretically or empirically. Here, using a strong predictor to fit the whole space could be treated as solving the predictor fitting subproblem relatively precisely, while adopting a weak predictor just imprecisely solves that. Previous methods solving Equation 2 truncate their solutions to "one shot" and hinge on solving subproblems with higher precision. Since we now take a joint optimization view and allow for multiple iterations, we can afford to only use weaker predictors for the fitting subproblem per iteration.

**Implementation Outline:** The above coordinate descent solution has clear interpretations and is straightforward to implement. Suppose our iterative methods has $K$ iterations. We initialize $S^1$ by randomly sampling a few samples from $X$, and train an initial predictor $\tilde{f}_1$. Then at iterations $k = 2, \ldots K$, we jointly optimize the sampling set $S^k$ and predictor $\tilde{f}_k$ in an alternative manner.

***Subproblem 1: Architecture Sampling.*** At iteration $k + 1$, we first sort all architectures[2] in the search space $X$ (excluding all the samples already in $S^k$) according to its predicted performance $\tilde{P}^k$ at

---

[2]One only exception is the Section 3.2 open-domain experiments: we will sub-sample all architectures in the search space before sorting. More details can be found in Appendix Section H

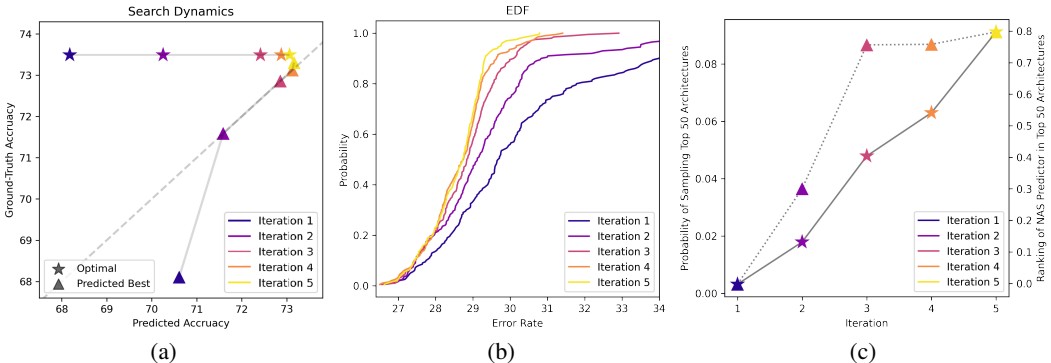

Figure 3: Visualization of the search dynamics in NAS-Bench-201 Search Space. (best viewed in color) (a) The trajectory of the predicted best architecture and global optimal through out 5 iterations; (b) Error *empirical distribution function* (EDF) of the predicted top-200 architectures throughout 5 iterations (c) Triangle marker: probability of sampling top-50 architectures throughout 5 iterations; Star marker: Kendall's Tau ranking of NAS predictor in Top 50 architectures through out 5 iterations.

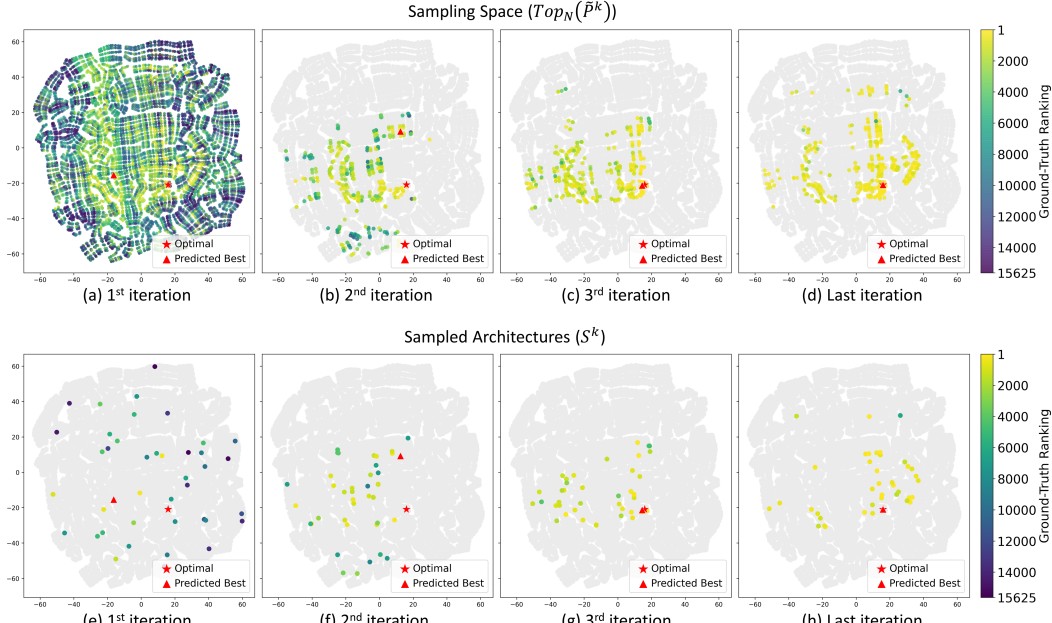

Figure 4: Visualization of search dynamics in NAS-Bench-201 Search Space via t-SNE. At $i$-th iteration, we randomly sample M = 40 new architectures from the top N = 400 ranked architectures in $\tilde{P}^k$. The top row from (a)-(d) show the sampling space $Top_N(\tilde{P}^k)$, and the bottom row from (e)-(h) show the sampled architectures $S^k$. The performance ranking of architectures is encoded by color, and those not-sampled architectures are colored in grey.

every iteration $k$. We then randomly sample $M$ new architectures from the top $N$ ranked architectures in $\tilde{P}^k$. Note this step both reduces the sample budget, and controls the exploitation-exploration trade-off (see Section 3.1). The newly sampled architectures together with $S^k$ become $S^{k+1}$.

***Subproblem 2: (Weak) Predictor Fitting.*** We learn a predictor $\tilde{f}^{k+1}$, by minimizing the loss $\mathcal{L}$ of the predictor $\tilde{f}^{k+1}$ based on sampled architectures $S^{k+1}$. We then evaluate architectures using the learned predictor $\tilde{f}^{k+1}$ to get the predicted performance $\tilde{P}^{k+1}$.

As illustrated in Figure 2, through alternating iterations, we progressively evolve weak predictors to focus on sampling along the search path, thus simplifying the learning workload of each predictor. With these coarse-to-fine iterations, the predictor $\tilde{f}^k$ would guide the sampling process to gradually zoom into the promising architecture samples. In addition, the promising samples $S^{k+1}$ would in turn improve the performance of the updated predictor $\tilde{f}^{k+1}$ among the well-performed architectures, hence the ranking of sampling space is also refined gradually. In other words, the solution quality to the subproblem 2 will gradually increase as a natural consequence of the guided zoom-in. For

derivation, we simply choose the best architecture predicted by the final weak predictor. This idea is related to the classical ensembling [25], yet a new regime to NAS.

***Proof-of-Concept Experiment.*** Figure 3 (a) shows the progressive procedure of finding the optimal architecture $x^*$ and learning the predicted best architecture $\tilde{x}_k^*$ over 5 iterations. As we can see from Figure 3 (a), the optimal architecture and the predicted best one are moving towards each other closer and closer, which indicates that the performance of predictor over the optimal architecture(s) is growing better. In Figure 3 (b), we use the error *empirical distribution function* (EDF) [26] to visualize the performance distribution of architectures in the subspace. We plot the EDF of the top-200 models based on the predicted performance over 5 iterations. As is shown, the subspace of top-performed architectures is consistently evolving towards more promising architecture samples over 5 iterations. Then in Figure 3 (c), we validate that the probabilities of sampling better architectures within the top $N$ predictions keep increasing. Based on this property, we can just sample a few well-performing architectures guided by the predictive model to estimate another better weak predictor. The same plot also suggests that the NAS predictor's ranking among the top-performed models is gradually refined, since more and more architectures in the top region are sampled.

In Figure 4, we also show the t-SNE visualization of the search dynamic in NAS-Bench-201 search space. We can observe that: (1) NAS-Bench-201 search space is highly structured; (2) the sampling space $Top_N(\tilde{P}^k)$ and sampled architectures $S^k$ are both consistently evolving towards more promising regions, as can be noticed by the increasingly warmer color trend.

### 2.3 Relationship to Bayesian Optimization: A Simplification and Why It Works

Our method can be alternatively regarded as a **vastly simplified variant** of Bayesian Optimization (BO). It does not refer to any explicit uncertainty-based modeling such as Gaussian Process (which are often difficult to scale up); instead it adopts a *very simple step function* as our acquisition function. For a sample $x$ in the search space $X$, our special "acquisition function" can be written as:

$$acq(x) = u(x - \theta) \cdot \epsilon \tag{5}$$

where the step function $u(x)$ is 1 if $x \geq \theta$, and 0 otherwise; $\epsilon$ is a random variable from the uniform distribution $U(0, 1)$; and $\theta$ is the threshold to split $Top N$ from the rest, according to their predicted performance $\tilde{P}^k(x)$. We then choose the samples with the $M$ largest acquisition values:

$$S_M = \underset{Top M}{\arg \max} \, acq(x) \tag{6}$$

*Why such "oversimplified BO" can be effectively for our framework?* We consider the reason to be the inherently structured NAS search space. Specifically, existing NAS spaces are created either by varying operators from a pre-defined operator set (DARTS/NAS-Bench-101/201 Search Space) or by varying kernel size, width or depth (MobileNet Search Space). Therefore, as shown in Figure 4, the search spaces are often highly-structured, and the best performers gather close to each other.

Here comes our underlying prior assumption: *we can progressively connect a piecewise search path from the initialization, to the finest subspace where the best architecture resides.* At the beginning, since the weak predictor only roughly fits the whole space, the sampling operation will be "noisier", inducing more exploration. When it comes to the later stage, the weak predictors fit better on the current well-performing clusters, thus performing more exploitation locally. Therefore our progressive weak predictor framework provides a natural evolution between exploration and exploitation, without explicit uncertainty modeling, thanks to the prior of special NAS space structure.

Another exploration-exploitation trade-off is implicitly built in the adaptive sampling step of our subproblem 1 solution. To recall, at each iteration, instead of choosing all Top $N$ models by the latest predictor, we randomly sample $M$ models from Top $N$ models to explore new architectures in a stochastic manner. By varying the ratio $\epsilon = M/N$ and the sampling strategy (e.g., uniform, linear-decay or exponential-decay), we can balance the sampling exploitation and exploration per step, in a similar flavor to the $\epsilon$-greedy [27] approach in reinforcement learning.

### 2.4 Our Framework is General to Predictor Models and Architecture Representations

Our framework is designed to be generalizable to various predictors and features. In predictor-based NAS, the objective of fitting the predictor $\tilde{f}$ is often cast as a regression [7] or ranking [5] problem. The choice of predictors is diverse, and usually critical to final performance [5, 6, 2, 7–9]. To illustrate our framework is generalizable and robust to the specific choice of predictors, we compare the following predictor variants.

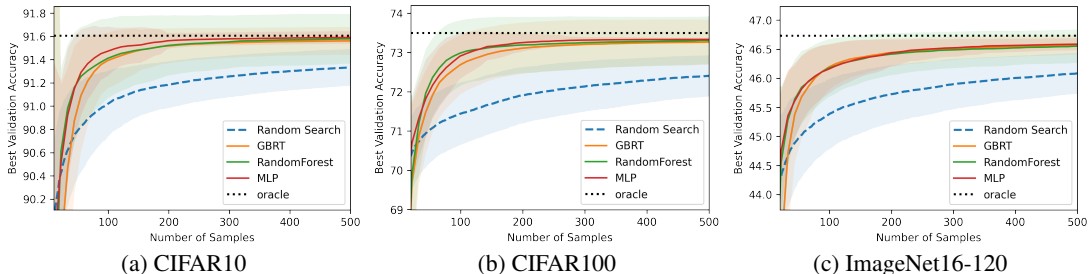

|  (a) CIFAR10 | (b) CIFAR100 | (c) ImageNet16-120 |

Figure 5: Evaluations of robustness across different predictors on NAS-Bench-201. Solid lines and shadow regions denote the mean and std, respectively.

- *Multilayer perceptron (MLP)*: MLP is the common baseline in predictor-based NAS [5] due to its simplicity. For our weak predictor, we use a 4-layer MLP with hidden layer dimension of (1000, 1000, 1000, 1000).
- *Regression Tree*: tree-based methods are also popular [9, 28] since they are suitable for categorical architecture representations. As our weak predictor, we use the Gradient Boosting Regression Tree (GBRT) based on XGBoost [29], consisting of 1000 Trees.
- *Random Forest*: random forests differ from GBRT in that they combines decisions only at the end rather than along the hierarchy, and are often more robust to noise. For each weak predictor, we use a random forest consisting of 1000 Forests.

The features representations to encode the architectures are also instrumental. Previous methods hand-craft various features for the best performance, e.g., raw architecture encoding [6], supernet statistics [30], and graph convolutional network encoding [7, 5, 8, 19] Our framework is also agnostic to various architecture representations, and we compare the following:

- *One-hot vector*: In NAS-Bench-201 [31], its DARTS-style search space has fixed graph connectivity, hence the one-hot vector is commonly used to encode the choice of operator.
- *Adjacency matrix*: In NAS-Bench-101, we used the same encoding scheme as in [32, 6], where a $7 \times 7$ adjacency matrix represents the graph connectivity and a 7-dimensional vector represents the choice of operator on every node.

As shown in Figure 5, all predictor models perform similarly across different datasets. Comparing performance on NAS-Bench-101 and NAS-Bench-201, although they use different architecture encoding methods, our method still performs similarly well among different predictors. This demonstrates that our framework is robust to various predictor and feature choices.

## 3 Experiments

**Setup:** For all experiments, we use an Intel Xeon E5-2650v4 CPU and a single Tesla P100 GPU, and use the Multilayer perceptron (MLP) as our default NAS predictor, unless otherwise specified.

**NAS-Bench-101** [32] provides a Directed Acyclic Graph (DAG) based cell structure. The connectivity of DAG can be arbitrary with a maximum number of 7 nodes and 9 edges. Each nodes on the DAG can choose from operator of $1 \times 1$ convolution, $3 \times 3$ convolution or $3 \times 3$ max-pooling. After removing duplicates, the dataset consists of 423,624 diverse architectures trained on CIFAR10[33].

**NAS-Bench-201** [31] is a more recent benchmark with a reduced DARTS-like search space. The DAG of each cell is fixed, and one can choose from 5 different operations ($1 \times 1$ convolution, $3 \times 3$ convolution, $3 \times 3$ avg-pooling, skip, no connection), on each of the 6 edges, totaling 15,625 architectures. It is trained on 3 different datasets: CIFAR10, CIFAR100 and ImageNet16-120 [34]. For experiments on both NAS-Benches, we followed the same setting as [8].

**Open Domain Search:** We follow the same NASNet search space used in [35] and MobileNet Search Space used in [36] to directly search for the best architectures on ImageNet[37]. Due to the huge computational cost to evaluate sampled architectures on ImageNet, we leverage a weight-sharing supernet approach. On NASNet search space, we use Single-Path One-shot [38] approach to train our SuperNet, while on MobileNet Search Space we reused the pre-trained supernet from OFA[36]. We then use the supernet accuracy as the performance proxy to train weak predictors. We clarify that despite using supernet, our method is more accurate than existing differentiable weight-sharing methods, meanwhile requiring less samples than evolution based weight-sharing

methods, as manifested in Table 6 and 7. We adopt PyTorch and image models library (timm) [39] to implement our models and conduct all ImageNet experiments using 8 Tesla V100 GPUs. For derived architecture, we follow a similar training from scratch strategies used in LaNAS[21].

## 3.1 Ablation Studies

We conduct a series of ablation studies on the effectiveness of proposed method on NAS-Bench-101. To validate the effectiveness of our iterative scheme, In Table 1, we initialize the initial Weak Predictor $\tilde{f}_1$ with 100 random samples, and set $M = 10$, after progressively adding more weak predictors (from 1 to 191), we find the performance keeps growing. This demonstrates the key property of our method that probability of sampling better architectures keeps increasing as more iteration goes. It's worth noting that the quality of random initial samples $M_0$ may also impact on the performance of WeakNAS, but if $|M_0|$ is sufficiently large, the chance of hitting good samples (or its neighborhood) is high, and empirically we found $|M_0|$=100 to already ensure highly stable performance at NAS-Bench-101: a more detailed ablation can be found in the Appendix.

| Sampling | #Predictor | #Queries | Test Acc.(%) | SD(%) | Test Regret(%) | Avg. Rank |
|---|---|---|---|---|---|---|
| Uniform | 1 Strong Predictor | 2000 | 93.92 | 0.08 | 0.40 | 135.0 |
| Iterative | 1 Weak Predictor | 100 | 93.42 | 0.37 | 0.90 | 6652.1 |
| | 11 Weak Predictors | 200 | 94.18 | 0.14 | 0.14 | 5.6 |
| | 91 Weak Predictors | 1000 | 94.25 | 0.04 | 0.07 | 1.7 |
| | 191 Weak Predictors | 2000 | 94.26 | 0.04 | 0.06 | 1.6 |
| Optimal | - | - | 94.32 | - | 0.00 | 1 |

Table 1: Ablation on the effectiveness of our iterative scheme on NAS-Bench-101

| Sampling (M from TopN) | M | TopN | #Queries | Test Acc.(%) | SD(%) | Test Regret(%) | Avg. Rank |
|---|---|---|---|---|---|---|---|
| Exponential-decay | 10 | 100 | 1000 | 93.96 | 0.10 | 0.36 | 85.0 |
| Linear-decay | 10 | 100 | 1000 | 94.06 | 0.08 | 0.26 | 26.1 |
| **Uniform** | **10** | **100** | **1000** | **94.25** | **0.04** | **0.07** | **1.7** |
| Uniform | 10 | 1000 | 1000 | 94.10 | 0.19 | 0.22 | 14.1 |
| **Uniform** | **10** | **100** | **1000** | **94.25** | **0.04** | **0.07** | **1.7** |
| Uniform | 10 | 10 | 1000 | 94.24 | 0.04 | 0.08 | 1.9 |

Table 2: Ablation on exploitation-exploration trade-off on NAS-Bench-101

| Method | #Queries | Test Acc.(%) | SD(%) | Test Regret(%) | Avg. Rank |
|---|---|---|---|---|---|
| **WeakNAS** | **1000** | **94.25** | **0.04** | **0.07** | **1.7** |
| WeakNAS (BO Variant) | 1000 | 94.12 | 0.15 | 0.20 | 8.7 |
| Optimal | - | 94.32 | - | 0.00 | 1.0 |

Table 3: Comparing to the BO variant of WeakNAS on NAS-Bench-101.

We then study the exploitation-exploration trade-off in Table 2 in NAS-Bench-101 (a similar ablation in Mobilenet Search space on ImageNet is also included in Appendix Table 6) by investigating two settings: (a) We gradually increase $N$ to allow for more exploration, similar to controlling $\epsilon$ in the epsilon-greedy [27] approach in the RL context; (b) We vary the sampling strategy from Uniform, Linear-decay to Exponential-decay (top models get higher probabilities by following either linear-decay or exponential-decay distribution). We empirically observed that: (a) The performance drops more (Test Regret 0.22% vs 0.08%) when more exploration (TopN=1000 vs TopN=10) is used. This indicates that extensive exploration is not optimal for NAS-Bench-101; (b) Uniform sampling method yields better performance than sampling method that biased

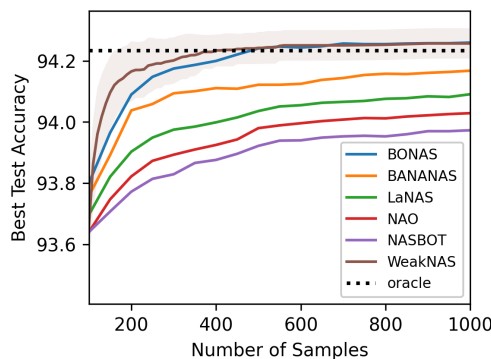

Figure 6: Comparison with SoTA methods on NAS-Bench-101. Solid lines and shadow regions denote the mean and std, respectively.

towards top performing model (e.g. linear-decay, exponential-decay). This indicates good architectures are evenly distributed within the Top 100 predictions of Weak NAS, therefore a simple uniform sampling strategy for exploration is more optimal in NAS-Bench-101. To conclude, our Weak NAS Predictor strikes a good balance between exploration and exploration.

Apart from the above exploitation-exploration trade-off of WeakNAS, we also explore the possibility of integrating other meta-sampling methods. We found that the local search algorithm could achieve comparable performance, while using Semi-NAS [20] as a meta sampling method could further boost the performance of WeakNAS: more details are in Appendix Section G.

## 3.2 Comparison to State-of-the-art (SOTA) Methods

**NAS-Bench-101:** On NAS-Bench-101 benchmark, we compare our method with several popular methods [14, 40, 21, 2, 7, 20, 19, 41–44]. Table 5 shows that our method significantly outperforms baselines in terms of sample efficiency. Specifically, our method costs $964\times$, $447\times$, $378\times$, $245\times$, $58\times$, and $7.5\times$ less samples to reach the optimal architecture, compared to Random Search, Regularized Evolution [14], MCTS [40], Semi-NAS[20], LaNAS[21], BONAS[19], respectively. We then plot the best accuracy against number of samples in Table 4 and Figure 6 to show the sample efficiency on the NAS-Bench-101, from which we can see that our method consistently costs fewer sample to reach higher accuracy.

| Method | #Queries | Test Acc.(%) | SD(%) | Test Regret(%) | Avg. Rank |
|---|---|---|---|---|---|
| Random Search | 2000 | 93.64 | 0.25 | 0.68 | 1750.0 |
| NAO [2] | 2000 | 93.90 | 0.03 | 0.42 | 168.1 |
| Reg Evolution [14] | 2000 | 93.96 | 0.05 | 0.36 | 85.0 |
| Semi-NAS [20] | 2000 | 94.02 | 0.05 | 0.30 | 42.1 |
| Neural Predictor [7] | 2000 | 94.04 | 0.05 | 0.28 | 33.5 |
| **WeakNAS** | **2000** | **94.26** | **0.04** | **0.06** | **1.6** |
| Semi-Assessor [42] | 1000 | 94.01 | - | 0.31 | 47.1 |
| LaNAS [21] | 1000 | 94.10 | - | 0.22 | 14.1 |
| BONAS [19] | 1000 | 94.22 | - | 0.10 | 3.0 |
| **WeakNAS** | **1000** | **94.25** | **0.04** | **0.07** | **1.7** |
| Arch2vec [41] | 400 | 94.10 | - | 0.22 | 14.1 |
| **WeakNAS** | **400** | **94.24** | **0.04** | **0.08** | **1.9** |
| LaNAS [21] | 200 | 93.90 | - | 0.42 | 168.1 |
| BONAS [19] | 200 | 94.09 | - | 0.23 | 18.0 |
| **WeakNAS** | **200** | **94.18** | **0.14** | **0.14** | **5.6** |
| NASBOWLr [45] | 150 | 94.09 | - | 0.23 | 18.0 |
| CATE (cate-DNGO-LS) [43] | 150 | 94.10 | - | 0.22 | 12.3 |
| **WeakNAS** | **150** | **94.10** | **0.19** | **0.22** | **12.3** |
| Optimal | - | 94.32 | - | 0.00 | 1.0 |

Table 4: Comparing searching efficiency by limiting the total query amounts on NAS-Bench-101.

| Method | NAS-Bench-101 | NAS-Bench-201 | | |
|---|---|---|---|---|
| Dataset | CIFAR10 | CIFAR10 | CIFAR100 | ImageNet16-120 |
| Random Search | 188139.8 | 7782.1 | 7621.2 | 7726.1 |
| Reg Evolution [14] | 87402.7 | 563.2 | 438.2 | 715.1 |
| MCTS [40] | 73977.2 | [†]528.3 | [†]405.4 | [†]578.2 |
| Semi-NAS [20] | [†]47932.3 | - | - | - |
| LaNAS [21] | 11390.7 | [†]247.1 | [†]187.5 | [†]292.4 |
| BONAS [19] | 1465.4 | - | - | - |
| **WeakNAS** | **195.2** | **182.1** | **78.4** | **268.4** |

Table 5: Comparison on the number of samples required to find the global optimal on NAS-Bench-101 and NAS-Bench-201. [†] denote reproduced results using adapted code.

**NAS-Bench-201:** We further evaluate on NAS-Bench-201, and compare with random search, Regularized Evolution [14], Semi-NAS[20], LaNAS[21], BONAS[19]. . As shown in Table 5, we conduct searches on all three subsets (CIFAR10, CIFAR100, ImageNet16-120) and report the average number

of samples needed to reach global optimal on the testing set over 100 runs. It shows that our method has the smallest sample cost among all settings.

**Open Domain Search:** we further apply our method to open domain search without ground-truth, and compare with several popular methods [35, 14, 46, 2, 47, 48, 21]. As shown in Tables 6 and 7, using the fewest samples (and only a fraction of GPU hours) among all, our method can achieve state-of-the-art ImageNet top-1 accuracies with comparable parameters and FLOPs. Our searched architecture is also competitive to expert-design networks. On the NASNet Search Space, compared with the SoTA predictor-based NAS method LaNAS (Oneshot) [21], our method reduces 0.6% top-1 error while using less GPU hours. On the MobileNet Search Space, we improve the previous SoTA LaNAS [21] to 81.3% top-1 accuracy on ImageNet while costing less FLOPs.

### 3.3 Discussion: Further Comparison with SOTA Predictor-based NAS Methods

**BO-based NAS methods [19, 45]**: BO-based methods in general treat NAS as a black-box optimization problem, for example, BONAS [19] customizes the classical BO framework in NAS with GCN embedding extractor and Bayesian Sigmoid Regression to acquire and select candidate architectures. The latest BO-based NAS approach, NASBOWL [45], combines the Weisfeiler-Lehman graph kernel in BO to capture the topological structures of the candidate architectures.

Compare with those BO-based method, our WeakNAS is an "oversimplified" version of BO as explained in Section 2.3. Interestingly, results in Table 4 suggests that WeakNAS is able to outperform BONAS [19], and is comparable to NASBOWLr [45] on NAS-Bench-101, showcasing that the simplification does not compromise NAS performance. We hypothesize that the following factors might be relevant: (1) the posterior modeling and uncertainty estimation in BO might be noisy; (2) the inherently structured NAS search space (shown in Figure 4) could enable a "shortcut" simplification to explore and exploit. In addition, the conventional uncertainty modeling in BO, such as the Gaussian Process used by [45], is not as scalable when the number of queries is large. In comparison, the complexity of WeakNAS scales almost linearly, as can be verified in Appendix Table 1. In our experiments, we observe WeakNAS to perform empirically more competitively than current BO-based NAS methods at larger query numbers, besides being way more efficient.

To further convince that WeakNAS is indeed an effective simplification compared to the explicit posterior modeling in BO, we report an apple-to-apple comparison, by use the same weak predictor from WeakNAS, plus obtaining its uncertainty estimation by calculating its variance using a deep ensemble of five model [49]; we then use the classic Expected Improvement (EI) [50] acquisition function. Table 3 confirms that such BO variant of WeakNAS is inferior our proposed formulation.

**Advanced Architecture Encoding [41, 43]** We also compare WeakNAS with NAS using custom architecture representation either in a unsupervised way such as arch2vec [41], or a supervised way such as CATE [43]. We show our WeakNAS could achieve comparable performance to both methods. Further, those architecture embedding are essentially **complementary** to our method to further boost the performance of WeakNAS, as shown in Appendix Section C.

**LaNAS [21]:** LaNAS and our framework both follow the divide-and-conquer idea, yet with two methodological differences: *(a) How to split the search space*: LaNAS learns a *classifier* to do binary "hard" partition on the search space (no ranking information utilized) and split it into two equally-sized subspaces. Ours uses a *regressor* to regress the performance of sampled architectures, and utilizes the ranking information to sample a percentage of the top samples ("soft" partition), with the sample size $N$ being controllable. *(b) How to do exploration*: LaNAS uses Upper Confidence Bound (UCB) to explore the search space by not always choosing the best subspace (left-most node) for sampling, while ours always chooses the best subspace and explore new architectures by adaptive sampling within it, via adjusting the ratio $\epsilon = M/N$ to randomly sample $M$ models from Top $N$. Tables 4 and 5 shows the superior sample efficiency of WeakNAS over LaNAS on NAS-Bench-101/201.

**Semi-NAS [20] and Semi-Assessor[42]:** Both our method and Semi-NAS/Semi-Assessor use an iterative algorithm containing prediction and sampling. The main difference lies in the use of pseudo labels: Semi-NAS and Semi-Assessor use pseudo labels as noisy labels to augment the training set, therefore being able to leverage "unlabeled samples" (e.g., architectures without true accuracies, but with only accuracies generated by the predictors) to update their predictors. Our method explores an **orthogonal** innovative direction, where the "pseudo labels" generated by the current predictor guide our sampling procedure, but are **never used** for training the next predictor.

That said, we point out that our method can be **complementary** to those semi-supervised methods [20, 42], thus they can further be **integrated** as one, For example, Semi-NAS can be used as a meta sampling method, where at each iteration we further train a Semi-NAS predictor with pseudo labeling strategy to augment the training set of our weak predictors. We show in Appendix Table 5 that the combination of our method with Semi-NAS can further boost the performance of WeakNAS.

**BRP-NAS [8]:** BRP-NAS uses a stronger GCN-based binary relation predictor which utilize extra topological prior, and leveraged a different scheme to control exploitation and exploration trade-off compare to our WeakNAS. Further, BRP-NAS also use a somehow unique setting, i.e. evaluating Top-40 predictions by the NAS predictor instead of the more common setting of Top-1 [2, 19, 21, 20]. Therefore, we include our comparison to BRP-NAS and more details in Appendix Section F.

| Model | Queries(#) | Top-1 Err.(%) | Top-5 Err.(%) | Params(M) | FLOPs(M) | GPU Days |
|---|---|---|---|---|---|---|
| MobileNetV2 | - | 25.3 | - | 6.9 | 585 | - |
| ShuffletNetV2 | - | 25.1 | - | 5.0 | 591 | - |
| SNAS[51] | - | 27.3 | 9.2 | 4.3 | 522 | 1.5 |
| DARTS[1] | - | 26.9 | 9.0 | 4.9 | 595 | 4.0 |
| P-DARTS[52] | - | 24.4 | 7.4 | 4.9 | 557 | 0.3 |
| PC-DARTS[53] | - | 24.2 | 7.3 | 5.3 | 597 | 3.8 |
| DS-NAS[53] | - | 24.2 | 7.3 | 5.3 | 597 | 10.4 |
| NASNet-A [35] | 20000 | 26.0 | 8.4 | 5.3 | 564 | 2000 |
| AmoebaNet-A [14] | 10000 | 25.5 | 8.0 | 5.1 | 555 | 3150 |
| PNAS [46] | 1160 | 25.8 | 8.1 | 5.1 | 588 | 200 |
| NAO [2] | 1000 | 24.5 | 7.8 | 6.5 | 590 | 200 |
| LaNAS [21] (Oneshot) | 800 | 24.1 | - | 5.4 | 567 | 3 |
| LaNAS [21] | 800 | 23.5 | - | 5.1 | 570 | 150 |
| **WeakNAS** | **800** | **23.5** | **6.8** | **5.5** | **591** | **2.5** |

Table 6: Comparison to SOTA results on ImageNet using NASNet search space.

| Model | Queries(#) | Top-1 Acc.(%) | Top-5 Acc.(%) | FLOPs(M) | GPU Days* |
|---|---|---|---|---|---|
| Proxyless NAS[54] | - | 75.1 | 92.9 | - | - |
| Semi-NAS[20] | 300 | 76.5 | 93.2 | 599 | - |
| BigNAS[47] | - | 76.5 | - | 586 | - |
| FBNetv3[48] | 20000 | 80.5 | 95.1 | 557 | - |
| OFA[36] | 16000 | 80.0 | - | 595 | 1.6 |
| LaNAS[21] | 800 | 80.8 | - | 598 | 0.3 |
| **WeakNAS** | **1000** | **81.3** | **95.1** | **560** | **0.16** |
| | **800** | **81.2** | **95.2** | **593** | **0.13** |

Table 7: Comparison to SOTA results on ImageNet using MobileNet search space. *Does not include supernet training cost.

## 4 Conclusions and Discussions of Broad Impact

In this paper, we present a novel predictor-based NAS framework named WeakNAS that progressively shrinks the sampling space, by learning a series of weak predictors that can connect towards the best architectures. By co-evolving the sampling stage and learning stage, our weak predictors can progressively evolve to sample towards the subspace of best architectures, thus greatly simplifying the learning task of each predictor. Extensive experiments on popular NAS benchmarks prove that the proposed method is both sample-efficient and robust to various combinations of predictors and architecture encoding means. However, WeakNAS is still limited by the human-designed encoding of neural architectures, and our future work plans to investigate how to jointly learn the predictor and encoding in our framework.

For broader impact, the excellent sample-efficiency of WeakNAS reduces the resource and energy consumption needed to search for efficient models, while still maintaining SoTA performance. That can effectively serve the goal of GreenAI, from model search to model deployment. It might meanwhile be subject to the potential abuse of searching for models serving malicious purposes.

## Acknowledgment

Z.W. is in part supported by an NSF CCRI project (#2016727).

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
