# Stronger NAS with Weaker Predictors
# Appendix

## A    Implementation details of baselines methods

For random search and regularized evolution[1] baseline, we use the public implementation from this link[1]. For random search, we selection 100 random architectures at each iteration. For regularized evolution, We set the initial population to 10, and the sample size each iteration to 3.

## B    Runtime comparsion of WeakNAS

We show the runtime comparison of WeakNAS and its BO variant in Table 1. We can see the BO variant is much slower in training proxy models due the ensembling of multiple models. Moreover, it's also several magnitude slower when deriving new samples, due to the calculation of its Expected Improvement (EI) acquisition function [2] being extremely costly.

| Method | Predictors | Config | Train proxy model (s/arch) | Derive new samples (s/arch) |
|---|---|---|---|---|
| WeakNAS | MLP | 4 layers @1000 hidden | $8.59 \times 10^{-5}$ | $3.53 \times 10^{-5}$ |
| | Gradient Boosting Tree | 1000 Trees | $5.70 \times 10^{-4}$ | $5.54 \times 10^{-7}$ |
| | Random Forrest | 1000 Forests | $3.20 \times 10^{-3}$ | $1.77 \times 10^{-4}$ |
| WeakNAS (BO Variant) | 5 x MLPs | EI acquisition | $2.84 \times 10^{-4}$ | $1.32 \times 10^{-1}$ |

Table 1: Runtime Comparsion of WeakNAS

## C    Ablation on the architecture encoding

We compare the effect of using different architecture encodings in in Table 2. We found when combined with CATE embedding [3], the performance of WeakNAS can be further improved, compared to WeakNAS baseline with adjacency matrix encoding used in [4]. This also leads to stronger performance than cate-DNGO-LS baseline in CATE [3], which demonstrates that CATE embedding [3] is an orthogonal contribution to WeakNAS, and they are mutually compatible.

| Methods | #Queries | Test Acc.(%) | SD(%) | Test Regret(%) | Avg. Rank |
|---|---|---|---|---|---|
| CATE (cate-DNGO-LS)[3] | 150 | 94.10 | - | 0.22 | 12.3 |
| WeakNAS + Adjacency matrix[4] | 150 | 94.10 | 0.19 | 0.22 | 12.3 |
| WeakNAS + CATE[3] | 150 | 94.19 | 0.12 | 0.13 | 5.24 |

Table 2: Details on Ablation on meta-sampling methods on NAS-Bench-101

## D    Ablation on number of initial samples

We conduct a controlled experiment in varying the number of initial samples $|M_0|$ in Table 3. On NAS-Bench-101, we vary $|M_0|$ from 10 to 200, and found a "warm start" with good initial samples is

---

[1]https://github.com/D-X-Y/AutoDL-Projects

crucial for good performance. Too small number of $|M_0|$ might makes the predictor lose track of the good performing regions. As shown in Table 3. We empirically found $|M_0|$=100 can ensure highly stable performance on NAS-Bench-101.

| $|M_0|$ | #Queries | Test Acc.(%) | SD(%) | Test Regret(%) | Avg. Rank |
|---------|----------|--------------|-------|----------------|-----------|
| 10 | 1000 | 94.14 | 0.10 | 0.18 | 9.1 |
| **100** | **1000** | **94.25** | **0.04** | **0.07** | **1.7** |
| 200 | 1000 | 94.19 | 0.08 | 0.13 | 5.2 |
| 10 | 200 | 94.04 | 0.13 | 0.28 | 33.5 |
| **100** | **200** | **94.18** | **0.14** | **0.14** | **5.6** |
| 200 | 200 | 93.78 | 1.45 | 0.54 | 558.0 |
| Optimal | - | 94.32 | - | 0.00 | 1.0 |

Table 3: Ablation on number of initial samples $M_0$ on NAS-Bench-101

# E    More comparison on NAS-Bench-201

We conduct a controlled experiment on NAS-Bench-201 by varying number of samples. As shown in Figure 1, our average performance over different number of samples is clearly better than Regularized Evolution [1] in all three subsets, with better stability indicated by confidence intervals.

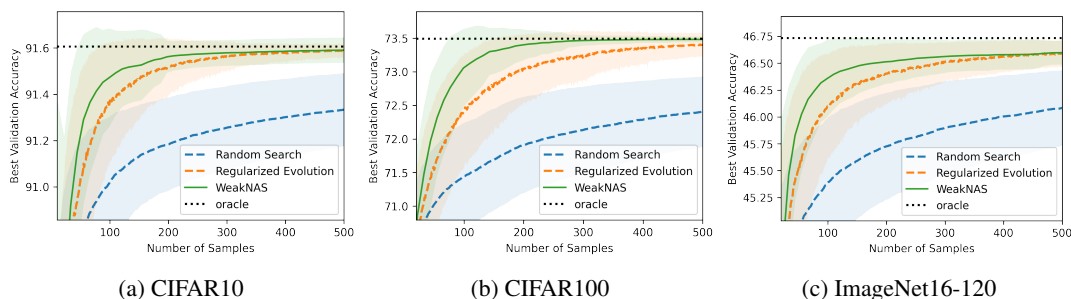

(a) CIFAR10                    (b) CIFAR100                    (c) ImageNet16-120

Figure 1: Comparison to SOTA on NAS-Bench-201 by varying number of samples. Solid lines and shadow regions denote the mean and std, respectively.

# F    Comparison to BRP-NAS

**Evaluation strategy**: BRP-NAS[5] uses a unique setting that differs from other predictor-based NAS, i.e., evaluating Top 40 predictions by the NAS predictor instead of Top 1 prediction, and the later was commonly followed by others[6–9] and WeakNAS.

**Sampling strategy**: WeakNAS uses a different sampling strategy than that of BRP-NAS, given a sample budget of $M$, BRP-NAS picks both samples from Top-$K$ and $(M - K)$ random models from the entire search space, while our WeakNAS only picks $M$ random models in Top-$N$, thus is a more "greedy" strategy. BRP-NAS controls the exploitation and exploration trade-off by adjusting $\alpha = (M - K)/M$, however they did not have any ablation discussing the exploitation and exploration trade-off and only empirically choose $\alpha = 0.5$ as the default ratio. Our WeakNAS instead controls the exploitation and exploration trade-off by adjusting $N/M$ ratio, and we did a comprehensive analysis on the exploitation and exploration trade-off on both NAS-Bench and MobileNet Search Space on ImageNet in Section **??**.

**NAS predictor**: BRP-NAS uses a stronger GCN-based binary relation predictors which utilizes extra topological prior, on the other hand, our framework generalizes to all choices of predictors, including MLP, Regression Tree and Random Forest, thus is not picky on the choice of predictors.

To fairly compare with BRP-NAS, we follow the exact same setting for our WeakNAS predictor, e.g., incorporating the same graph convolutional network (GCN) based predictor and using Top-40 evaluation. As shown in Table 4, at 100 training samples, WeakNAS can achieve comparable performance to BRP-NAS [5].

| Method | #Train | #Queries | Test Acc.(%) | SD(%) | Test Regret(%) | Avg. Rank |
|---|---|---|---|---|---|---|
| BRP-NAS [5] | 100 | 140 | 94.22 | - | 0.10 | 3.0 |
| **WeakNAS** | **100** | **140** | **94.23** | **0.09** | **0.09** | **2.3** |
| Optimal | - | - | 94.32 | - | 0.00 | 1.0 |

Table 4: Comparison to BRP-NAS on NAS-Bench-101.

## G   Comparsion of meta-sampling methods in WeakNAS

We also show that local search algorithm (hill climbing) or Semi-NAS [9] can be used as a meta sampling method in WeakNAS, which could further boost the performance of WeakNAS, here are the implementation details.

**Local Search** Given a network architecture embedding $s$ in NAS-Bench-101 Search Space, we first define a nearest neighbour function $N(s)$ as architecture that differ from $s$ by a edge or a operation. At each iteration, we random sample a initial sample $s_i$ from TopN predictions $\text{Top}_N(\tilde{P}^k)$ and sample all of its nearest neighbour architecture in $N(v_0)$. We then let the new $s_{i+1} = \arg\max_{s \in N(s_i)} f(s)$. We repeat the process iteratively until we reach a local maximum such that $\forall v \in N(s), f(s) \geqslant f(v)$ or the sampling budget $M$ of the iteration is reached.

**Semi-NAS** At the sampling stage of each iteration in WeakNAS, we further use Semi-NAS as a meta-sampling methods. Given a meta search space of 1000 architectures and a sample budget of 100 queries each iteration. We following the setting in Semi-NAS, using the same 4-layer MLP NAS predictor in WeakNAS and uses pseudo labels as noisy labels to augment the training set, therefore we are able to leverage "unlabeled samples" (e.g., architectures with accuracy generated by the predictors) to update the predictor. We set the initial sample to be 10, and sample 10 more samples each iteration. Note that at the start of $k$-th WeakNAS iteration, we inherent the weight of Semi-NAS predictor from the previous $(k$-1)-th WeakNAS iteration.

| Sampling (M from TopN) | M | N | #Queries | Test Acc.(%) | SD(%) | Test Regret(%) | Avg. Rank |
|---|---|---|---|---|---|---|---|
| WeakNAS | 100 | 1000 | 1000 | 94.25 | 0.04 | 0.07 | 1.7 |
| Local Search | - | - | 1000 | 94.24 | 0.03 | 0.08 | 1.9 |
| Semi-NAS | - | - | 1000 | 94.26 | 0.02 | 0.06 | 1.6 |

Table 5: Ablation on meta-sampling methods on NAS-Bench-101

## H   Details of Implementation on Open Domain Search Space

We extend WeakNAS to open domain settings by (a) Construct the evaluation pool $\bar{X}$ by uniform sampling the whole search space $X$ (b) Apply WeakNAS in the evaluation space $\bar{X}$ to find the best performer. (c) Train the best performer architecture from scratch.

For instance, when working with MobileNet search space that includes $\approx 10^{18}$ architectures, we uniformly sample 10K models as an evaluation pool, and further apply WeakNAS with a sample budget of 800 or 1000. When working with NASNet search space that includes $\approx 10^{21}$ architectures, we uniformly sample 100K models as an evaluation pool, and further apply WeakNAS with a sample budget of 800.

In the following part, we take MobileNet open domain search space as a example, however we follow a similar procedure for NASNet search space.

**(a) Construct the evaluation pool $\bar{X}$ from the search space $X$** We uniformly sample an evaluation pool to handle the extremely large MobileNet search space ($|X| \approx 10^{18}$), since its not doable to predict the performance of all architectures in $X$. We use uniform sampling due to a recent study [10] reveal that human-designed NAS search spaces usually contain a fair proportion of good models compared to random design spaces, for example, in Figure 9 of [10], it shows that in NASNet/Amoeba/PNAS/ENAS/DARTS search spaces, Top 5% of models only have a <1% performance gap to the global optima. In practice, the uniform sampling strategy has been widely

verified as effective in other works of predictor-based NAS such as [11–13], For example, [11] [12][13] set to be 112K, 15K, 20K in a search space of $10^{18}$ networks. In our case, we set $|\bar{X}| = 10K$.

**(b) Apply WeakNAS in the evaluation space $\bar{X}$** We then further apply WeakNAS in the evaluation pool $\bar{X}$. This is because even with the evaluation pool $|\bar{X}| = 10K$, it still takes days to evaluate all those models on ImageNet (in a weight-sharing SuperNet setting). Since the evaluation pool $\bar{X}$ was uniformly sampled from NAS search space $X$, it preserves the highly-structured nature of $X$. As a result, we can leverage WeakNAS to navigate through the highly-structured search space. WeakNAS build a iterative process, where it searches for some top-performing cluster at the initial search iteration and then "zoom-in" the cluster to find the top performers within the same cluster (as shown in Figure **??**). At $k - th$ iteration, WeakNAS balance the exploration and exploitation trade-off by sampling 100 models from the Top 1000 predictions of the predictor $\tilde{f}^k$, it use the promising samples to further improve performance of the predictor in the next iteration $\tilde{f}^{k+1}$. We leverage WeakNAS to further decrease the number of queries to find the optimal in $\bar{X}$ by 10 times, the search cost has dropped from 25 GPU hours (evaluate all 10K samples in random evaluation pool) to 2.5 GPU hours (use WeakNAS in 10K random evaluation pool), while still achieving a solid performance of 81.3% on ImageNet (MobileNet Search Space).

**(c) Train the best performer architecture from scratch.** We follow a similar setting in LaNAS[8], where we use Random Erase and RandAug, a drop out rate of 0.3 and a drop path rate of 0.0, we also use exponential moving average (EMA) with a decay rate of 0.9999. During training and evaluation, we set the image size to be 236x236 (In NASNet search space, we set the image size to be 224x224). We train for 300 epochs with warm-up of 3 epochs, we use a batch size of 1024 and RMSprop as the optimizer. We use a cosine decay learning rate scheduler with a starting learning rate of 1e-02 and a terminal learning rate of 1e-05.

# I   Ablation on exploitation-exploration trade-off in Mobilenet Search space on ImageNet

For the ablation on open-domain search space, we follow the same setting in the Section H, however due to the prohibitive cost of training model from scratch in Section H (c), we directly use accuracy derived from supernet.

WeakNAS uniformly samples M samples from TopN predictions at each iteration, thus we can adjust N/M ratio to balance the exploitation-exploration trade-off. In Table 6, we set the total number of queries at 100, fix $M$ at 10 and while adjusting $N$ from 10 (more exploitation) to 1000 (more exploration), and use optimal in the 10K evaluation pool to measure the ranking and test regret. We found WeakNAS is quite robust within the range where N/M = 2.5 - 10, achieving the best performance at the sweet spot of N/M = 5. However, its performance drops significantly (by rank), while doing either too much exploitation (N/M <2.5) or too much exploration (N/M >25).

| Sampling methods | M | TopN | #Queries | SuperNet Test Acc.(%) | SD(%) | Test Regret(%) | Avg. Rank |
|---|---|---|---|---|---|---|---|
| Uniform | - | - | 100 | 79.0609 | 0.0690 | 0.1671 | 94.58 |
| Iterative | 10 | 10 | 100 | 79.1552 | 0.0553 | 0.0728 | 20.69 |
| | 10 | 25 | 100 | 79.1936 | 0.0289 | 0.0344 | 4.68 |
| | **10** | **50** | **100** | **79.2005** | **0.0300** | **0.0275** | **4.05** |
| | 10 | 100 | 100 | 79.1954 | 0.0300 | 0.0326 | 4.63 |
| | 10 | 250 | 100 | 79.1755 | 0.0416 | 0.0525 | 10.58 |
| | 10 | 500 | 100 | 79.1710 | 0.0388 | 0.0570 | 10.80 |
| | 10 | 1000 | 100 | 79.1480 | 0.0459 | 0.0800 | 19.70 |
| | 10 | 2500 | 100 | 79.1274 | 0.0597 | 0.1006 | 33.64 |

Table 6: Ablation on exploitation-exploration trade-off over 100 runs on MobleNet Search Space over ImageNet

# J   Founded Architecture on Open Domain Search

We show the best architecture founded by WeakNAS with 800/1000 queries in Table 7.

| Id | Block | Kernel | #Out Channel | Expand Ratio | Id | Block | Kernel | #Out Channel | Expand Ratio |
|---|---|---|---|---|---|---|---|---|---|
| WeakNAS @ 593 MFLOPs, #Queries=800 | | | | | WeakNAS @ 560 MFLOPs, #Queries=1000 | | | | |
| 0 | Conv | 3 | 24 | - | 0 | Conv | 3 | 24 | - |
| 1 | IRB | 3 | 24 | 1 | 1 | IRB | 3 | 24 | 1 |
| 2 | IRB | 3 | 32 | 4 | 2 | IRB | 5 | 32 | 3 |
| 3 | IRB | 5 | 32 | 6 | 3 | IRB | 3 | 32 | 3 |
| 4 | IRB | 7 | 48 | 4 | 4 | IRB | 3 | 32 | 4 |
| 5 | IRB | 5 | 48 | 3 | 5 | IRB | 3 | 32 | 3 |
| 6 | IRB | 7 | 48 | 4 | 6 | IRB | 5 | 48 | 4 |
| 7 | IRB | 3 | 48 | 6 | 7 | IRB | 5 | 48 | 6 |
| 8 | IRB | 3 | 96 | 4 | 8 | IRB | 5 | 48 | 4 |
| 9 | IRB | 7 | 96 | 6 | 9 | IRB | 7 | 96 | 4 |
| 10 | IRB | 5 | 96 | 6 | 10 | IRB | 5 | 96 | 6 |
| 11 | IRB | 7 | 96 | 3 | 11 | IRB | 7 | 96 | 6 |
| 12 | IRB | 3 | 136 | 6 | 12 | IRB | 3 | 136 | 6 |
| 13 | IRB | 3 | 136 | 6 | 13 | IRB | 5 | 136 | 6 |
| 14 | IRB | 5 | 136 | 6 | 14 | IRB | 5 | 136 | 6 |
| 15 | IRB | 5 | 136 | 3 | 15 | IRB | 7 | 192 | 6 |
| 16 | IRB | 7 | 192 | 6 | 16 | IRB | 5 | 192 | 6 |
| 17 | IRB | 5 | 192 | 6 | 17 | IRB | 3 | 192 | 6 |
| 18 | IRB | 3 | 192 | 4 | 18 | IRB | 5 | 192 | 3 |
| 19 | IRB | 5 | 192 | 3 | 19 | Conv | 1 | 192 | - |
| 20 | Conv | 1 | 192 | - | 20 | Conv | 1 | 1152 | - |
| 21 | Conv | 1 | 1152 | - | 21 | FC | - | 1536 | - |
| 22 | FC | - | 1536 | - | | | | | |

Table 7: Neural architecture found by WeakNAS on ImageNet using MobileNet search space, i.e. results in main paper Table 6