# OpenReview forum: "Stronger NAS with Weaker Predictors"
_NeurIPS.cc/2021/Conference — NeurIPS 2021 Poster_

### Official Review · Reviewer_eYgg · 2021-07-05

**Rating:** 6
**Confidence:** 3

**Summary:**

Most Neural Architecture Search (NAS) techniques rely on being able to predict the performance of a network on a task without the cost of a full training run. This paper breaks down the methods of prediction into two categories: strong predictors and weak predictors. The authors note that this assumes that we want to evaluate all samples from the search space uniformly, even if we know certain subspaces are less interesting. Overall I think this is an interesting insight and in itself quite novel.

They therefore propose a method that attempts to improve the sample efficiency of NAS by using a series of “weaker” predictors to identify interesting subspaces, and then sample from those subspaces.

**Ethical Concerns:**

I have no ethical concerns with this paper.

**Limitations And Societal Impact:**

The authors have adequately potential negative societal impact. Please see the main review above for my questions.

**Main Review:**

Overall I think this paper has fallen foul of unfortunate timing, since it seems like a nice idea which has potentially become less relevant in the last year with the advent of zero-cost proxies. For me, at least a comparison to [1] or discussion of how the methods compare is essential for this paper to be accepted.

Otherwise, the paper is well written and the narrative is easy to follow but I found it quite hard to glean the specific implementation details. The idea that we can use weak predictors to select subspaces is appealing, but the mechanics of actually what that predictor looks like, what it’s predicting, and how that fits into the framework was quite hard for me to figure out.

My understanding is as follows: the typical setup here is that we have an MLP (or many MLPs?) which is going to predict a performance value for each network. We iterate between: sampling the top-N scoring networks from our predictions, and fitting the predictor

I have a few questions:
1. Do we have to start by first predicting a value for all of the networks in the space?
2. We sample the top-N scoring networks from the predictions and fit the predictor on them. Then we “evaluate” the architectures using the predictor. What does this mean? Haven’t you just trained the networks in order to fit the predictor? Doesn’t it make more sense to just use the accuracies of the trained networks directly?
3. On the next iteration, do we sample from the whole search space again or do we just sample from the top-N architectures we ranked last iteration?
4. If we have multiple predictors, are they sampling different parts of the space? Or are they both being trained on exactly the same data?

The authors write out predictor-based NAS as a bi-level optimization problem (sampling + predictor fitting), which feels obvious at first but actually makes for quite a nice general framework: for example, we see that predictor-based NAS methods are really doing co-ordinate descent, which is something I had never really thought of before. As a consequence of viewing it this way, the idea of having weak predictors starts to make a lot more sense, which is really nice for the reader to follow along with.

## Positive points
- The use of figures is nice and illustrative, and makes the narrative of the paper clear and easy to read
- The results are good and don’t seem cherry picked — all of the baselines/architectures look very sensible (e.g. using 1000 forests, 1000 trees, 1000 hidden units). The comparison to random search is very welcome, and the authors include error bars over 50 runs which is good to see
- Using average rank + test regret as evaluation measures makes for a compelling case

## Points to address
5. I think the authors should be a lot more upfront about what a strong/weak predictor is (i.e. within the first page we should have some idea of what is a strong vs a weak predictor)
6. I’m a little confused by the proof-of-concept experiment in section 2.2. What is the predictor being used? In figure 3(a) I can’t wrap my head around why the predicted accuracy of the global optimal point moves. Can the authors clarify this?
7. I know that the papers were released at very similar times, but the paper feels incomplete without a reference/comparison to [1], which proposes a range of extremely cheap predictors. From the results in tables 1 and 3 it seems like this method will be stronger (and indeed it seems like it might be possible to use the two approaches in conjunction). Some discussion on this or a convincing comparison would go a long way to improving the paper.

**Originality** Addressing the expense of neural architecture search is not particularly "new" or "novel", but I do think the idea to break the problem down into subsampling tasks is nice and not something I have seen before.   Differentiation between this work and previous contributions is clear. Related work is mostly adequately cited, though the paper is missing comparisons to the various "zero-cost" estimators listed in [1]

**Quality:** The submission seems technically sound, with a nice logical flow to the experiments. I enjoyed reading the paper and seeing the figures/tables to support the claims being made in the paper. Each claim seems well-justified.

**Clarity:** The submissions is relatively clear but I did get a little confused by the details of the sampling/resampling process (see my above questions). I'm not 100% sure I could reproduce the method from its current description.

**Significance:** The importance of the results may be slightly diminished by the development of work on cheap estimators for neural architecture search (though I do also believe the authors could address this). Otherwise, the results are nice.

## Minor typographical notes:
- Capitalization on subsection 2.2 is strange (why is A capitalized?)
- Figure 3 (a) accuracy is misspelt
- “Why such “oversimplified BO" can be effectively for our framework?“ -> Why can such “oversimplified” BO be effective for our framework?

[1] Mohamed S Abdelfattah, Abhinav Mehrotra, Łukasz Dudziak, and Nicholas Donald Lane. Zero-cost proxies for lightweight nas. In Proceedings of the International Conference on Learning Representations (ICLR), 2021.

**Time Spent Reviewing:**

5

---

> ### Author Response · Authors · 2021-08-10
> **Respond to Reviewer eYgg**
>
> Q1: Do we have to start by first predicting a value for all of the networks in the space?
> - For the first iteration, we start from random sampling a few architectures instead of predicting performance for all architectures.
>
> Q2: We sample the top-N scoring networks from the predictions and fit the predictor on them. Then we “evaluate” the architectures using the predictor. What does this mean? Haven’t you just trained the networks in order to fit the predictor? Doesn’t it make more sense to just use the accuracies of the trained networks directly?
> - We first sample the top-N scoring networks from the predictions of the previous NAS predictor, then we train a predictor using those samples, and we evaluate all the architectures in the search space with that predictor to get “predicted accuracies”. After that, we sample the top-N scoring networks from the predicted accuracies and repeat the same process. We only use the “predicted” accuracies to rank all the architectures, since they are all normalized at the same scale, if we replace part of the already sampled architectures with the ground truth accuracy, the ranking would be inaccurate.
>
> Q3: On the next iteration, do we sample from the whole search space again or do we just sample from the top-N architectures we ranked last iteration?
> - We just sample from the top-N architectures we ranked the last iteration, this is how we gradually refine the sampling space.
>
> Q4: If we have multiple predictors, are they sampling different parts of the space? Or are they both being trained on exactly the same data?
> - If you are talking about multiple predictors across different iterations, the first predictor will sample from the whole search space, while the latter predictors would sample from a gradually refined search space, according to the prediction of the previous predictor, so essentially multiple predictors are working on a different part of the search space while being more and more confined.
>
> Q5: I think the authors should be a lot more upfront about what a strong/weak predictor is (i.e. within the first page we should have some idea of what is a strong vs a weak predictor)
> - We want to clarify that we named “weak predictor” because it only predicts a local subspace of the search space thus is associated with our iterative sampling scheme (and therefore it usually does not need very heavily parameterized models). On the contrary, previous “strong predictor” predicts the global search space and is associated with uniform sampling. It does not represent the number of parameters or type of NAS predictor (e.g. MLP, GBRT, GCN, Random Forrest) as they are loose indicators for representation capacity. A very heavily parameterized NAS predictor with our iterative sampling scheme is still consider “weak predictors”.
>
> Q6: I’m a little confused by the proof-of-concept experiment in section 2.2. What is the predictor being used? In figure 3(a) I can’t wrap my head around why the predicted accuracy of the global optimal point moves. Can the authors clarify this?
> - The predictor that has been used is a 4-layer Multilayer perceptron (MLP) with hidden dimension of 1000 as described in section 2.4.
> This is because as the iterative process moves on, the predictor itself is also evolving due to more and more samples coming in, so the prediction of the same global optimal architecture would also change accordingly. We will clarify in the revised paper.
>
> Q7: I know that the papers were released at very similar times, but the paper feels incomplete without a reference/comparison to [1], which proposes a range of extremely cheap predictors. From the results in tables 1 and 3 it seems like this method will be stronger (and indeed it seems like it might be possible to use the two approaches in conjunction). Some discussion on this or a convincing comparison would go a long way to improving the paper.
> - Thanks for pointing out the reference! Here we attach the comparison to [1] on the NAS-Bench-101 Validation Set in the following Table
> We also agree the two methods can be applied together, and we are working on the comparison with [1] now (will update once ready).
> - | Method |  \#Queries  | Test Acc.(\%) | SD(\%) | Test Regret(\%) | Avg. Rank |
> |:----------|:----------:|------------:|------------:|----------:|---------:|
> | Zero-proxy (RAND + warmup (15000)) [1] | 150| 94.40 | - | 0.65 | 384.0 |
> | Zero-proxy (AE + warmup (15000)) [1] | 150 | 94.49 | - | 0.56 | 157.3 |
> | **WeakNAS** |**150** | **94.70** | **0.06** | **0.35** | **13.8** |
>
> Q8: Minor typos and clarity
> - Thanks for pointing out those typos, we will fix them in the final version. Please also be assured that all our codes will be publicly available on GitHub for 100% reproducibility.

---

> > ### Comment · Reviewer_eYgg · 2021-08-14
> > **Response to authors**
> >
> > Thanks for the clarifications.
> >
> > I think there is one piece I am still misunderstanding.
> >
> > > then we train a predictor using those samples, and we evaluate all the architectures in the search space with that predictor to get “predicted accuracies”
> >
> > In particular, "then we train a predictor using those samples" --- I don't understand what the "train" step is here. What are your inputs and targets?
> >
> > =================
> >
> > With the inclusion of the Q7 results (plus ideally a few more results along the same lines) I'm happy to raise my score.

---

> > > ### Author Response · Authors · 2021-08-16
> > > **Response to reviewer**
> > >
> > > Thanks for your response!
> > >
> > > Q1: "then we train a predictor using those samples"  I don't understand what the "train" step is here. What are your inputs and targets?
> > >
> > > - In WeakNAS, we train the NAS predictor, which is a function that uses network embedding as input, and outputs predicted accuracy. The training data includes all sampled networks since the first iteration. For each network, the input is the network embedding, and the target is the ground truth accuracy, which is obtained by training the corresponding network from scratch.
> > >
> > > Q2: Inclusion of the Q7 results
> > >
> > > - We are already working on the zero-cost proxies + WeakNAS experiment and should get back to you once it's completed.

---

### Official Review · Reviewer_YANc · 2021-07-15

**Rating:** 6
**Confidence:** 4

**Summary:**

This paper proposes a simple approach to predictor-based neural architecture search. It is motivated by the observation that in NAS, we are really just interested in precisely estimating the performance of well performing architectures while less accurate estimates are tolerable for low performing architectures (as long as the predictor facilitates to discriminate between the two). Therefore the paper proposes to learn a series of performance predictors that subsequently focus on better performing architectures, where each new predictor relies of the previous one, instead of learning a single predictor in one step.
In each step, a new sample of predicted to be well performing architectures is drawn and evaluated. The experiments show that this leads to very efficient sampling of increasingly good architectures on several search spaces inclusing NasBench 101 and NasBench201, NatNet and MobileNet Search spaces from the literature such that well performing architectures can be found with few samples.


**Limitations And Societal Impact:**

With respect to the limitations, it would be good to also discuss the assumptions implicitly made on the regularity of the search space (see above). Otherwise, limitations and broader impact are appropriately addressed.

**Main Review:**

The idea proposed in this paper is very simple. Yet, it seems to be efficient on the considered search spaces. I would expect an additional discussion on the search spaces in this respect: What assumptions do we need to make on the search space? Under which conditions would well performing architectures be discarded early on? In my understanding, this approach should work on convex search spaces and it is likely  to discard narrow optima in non-convex ones. Is that correct?

Besides the simplicity and practical effectiveness of of the idea, further strengths of the paper are:

- The paper/idea is well motivated.
- The mathematical formalization makes sense.
- The approach is evaluated for several predictors and datasets.
- An ablation study in the number of query samples is provided.
- An ablation study on the new hyperparameters M and N is provided.
- The evaluation on the NasBench101 and 201 search spaces is thorough.

Weaknesses and questions:

- The method introduces new hyperparameters that need ti be optimzed per search space.
- In the ablation study on the number of samples, the chosen #of predictors is a bit unusual (6, 11, 91 predictors, etc).
- What is the training time per predictor?
- The Improvement over BONAS with respect to Test acc. for a give query budget is very small on NasBench101 (Table.  3)
- It would be interesting to see the same experiment as in Table 3 for NasBench 201.
- All number in tables 1 - 5 are already given as average values over 25 runs. The standard deviation should be reported as well.
- In Figure 5, the comparison to BONAS would be very helpful.


The paper is overall well written and well structured. Yet, there are several minor language errors that should be addressed, for example:
line 56: "denoted" should be "denote"
line 105: "just imprecisely solves that" please reword
line 105: "previous methods" - please specify which methods you refer to
line 147: "can be effectively for our framework" should be "can be effective"
line 202: "choose from operator of" please reword
line 226: ", In"
line 231: "its" should be "their"
These are minor issues since the understanding is not impaired, but they should be fixed.

Similarly, in Equation (3), please check the \setminus. This does not seem to be correct. I think is should be \subseteq







**Time Spent Reviewing:**

4h

---

> ### Author Response · Authors · 2021-08-10
> **Respond to Reviewer YANc**
>
> Q1: What assumptions do we need to make on the search space? Under which conditions would well-performing architectures be discarded early on? In my understanding, this approach should work on convex search spaces and it is likely to discard narrow optima in non-convex ones. Is that correct?
> - This is a great question for discussion. WeakNAS indeed works the best on the convex space and non-convex spaces with flatter optima. Even on non-convex one with the narrow optima, it still has chances as long as we sample within/near the narrow optima. During the iterative process, once any of the samples drop into the (close) neighborhood of narrow optima, we can still quickly converge into the optima.
> - It is a challenging task to guarantee to find the narrow optima in a non-convex space. In practice, we evaluate existing NAS search space and are able to consistently find good architecture in NAS-Bench-101, NAS-Bench-201, NASNet, and MobileNet Search Space, although those are practically complicated search spaces.
>
> Q2: The method introduces new hyperparameters that need to be optimized per search space.
> - Our method indeed introduced extra hyperparameters to tune, however, we include an ablation on the choice of sampling hyperparameters in Table. 2. In general, our method is NOT sensitive to the choice of M and N, hence no much overhead to tune them is incurred. As a rule of thumb, on small search spaces such as NAS-Bench-101/201 with samples < 500K, we choose small M=10, on large open-domain search spaces such as NASNet or MobileNet with samples >10^10, we choose M=100.
>
> Q3: In the ablation study on the number of samples, the chosen #of predictors is a bit unusual (6, 11, 91 predictors, etc).
> - This is because we initialize the predictor with 100 samples and add 10 more samples each iteration. So that if we use #Queries=100 would have 1 weak predictor,  #Queries=150 would have 1+5=6 weak predictors, #Queries=1000 would have 1+90=91 weak predictors. We will clarify this in the revised paper.
> Q4: What is the training time per predictor?
> - We evaluate the runtime of our NAS predictor using a Xeon E5-2650V4 CPU, shown in the table below
>
> - | Weak NAS Predictors | Config | train proxy model (s/arch) | eval proxy model (s/arch) |
> |:----------|:-----------|:----------:|:------------:|
> | MLP | 4 layers @1000 hidden| $8.59 ×10^{-5}$  | $3.53×10^{-5}$ |
> | Gradient Boosting Tree | 1000 Trees             | $5.70 ×10^{-4}$ | $5.54 ×10^{-7}$ |
> | Random Forrest |1000 Forrests                     | $3.20 ×10^{-3}$ | $1.77 ×10^{-4}$ |
> - We can see the runtime of the predictor is extremely low.
>
> Q5: The Improvement over BONAS with respect to Test acc. for a given query budget is very small on NasBench101 (Table. 3)
> - Note that the goal of NAS is to find the closest architectures to the optimal, and currently the ranking is a more important metric in NAS. Our WeakNAS is significantly better at predicting ranking of top architectures thanks to the fine-grained modeling focused on the best architecture regions.
> - Despite the accuracy gap of 0.1%, the distance to the optimal (or test regret) improved from 0.23 by 40% to 0.14, while the average rank is 3.2 times better than BONAS, respectively, at 200 queries, This is a very notable and clear gain.
> Furthermore, Table 4/5/6 also shows our advantage over BONAS is highly consistent, on NAS-Bench-201 all 3 subsets, NASNet Search Space, MobileNet Search space on ImageNet. That supplies further evidence of statistical significance.
>
> Q6: It would be interesting to see the same experiment as in Table 3 for NasBench 201.
> - We run the requested experiment and report result on NASBench201 CIFAR10 subset, in the following table:
>
> - | NAS Predictor |  \#Queries  | Test Acc.(\%) | SD(\%) | Test Regret(\%) |
> |:----------|----------:|------------:|------------:|----------:|
> | Random | 200| 91.21 | 0.20 | 0.41 |
> | Regulazied Evolution | 200 | 91.50 | 0.14  | 0.12 |
> | LaNAS | 200 | 91.52 | 0.12 | 0.10 |
> | WeakNAS | 200 | 91.58 | 0.08 | 0.04 |
>
> Q7: The standard deviation should be reported as well.
> - The standard deviation has already been reported in the “SD(%)” column of Tables 1-4. We will make it clear in the final version.
>
> Q8: In Figure 5, the comparison to BONAS would be very helpful.
> - Yes, we agree. As rebuttal won’t allow us to update figures, we promise to add the comparison to BONAS on Figure 5 in the final version. We already have the curves and they are aligned with all current observations/conclusions.
>
> Q9: Minor typos
> - Thanks for pointing out those typos! We will fix them in the final version.

---

> > ### Comment · Reviewer_YANc · 2021-08-19
> > **Thanks for the clarifications.**
> >
> > Thanks for the clarifications. It would be great to include them in the final version.

---

### Official Review · Reviewer_onF1 · 2021-07-16

**Rating:** 7
**Confidence:** 5

**Summary:**


This paper proposes a new NAS method, which is a crucial step to rescue the NAS community. This paper achieves state-of-the-art results on several benchmarks, especially Open Domain Search on ImageNet. Sufficient experiments have been conducted to prove the effectiveness of the proposed method. Proper principled analyses have been provided to show the validness of the proposed method.



**Ethical Concerns:**


No.



**Limitations And Societal Impact:**



Please refer to and respond to the "negative" items in the above main review.



**Main Review:**



(Positive) I am really convinced by the motivation of this paper, especially by the impressive curve in Figure 1.

(Weakness) The motivation section needs to be improved. In the first paragraph, the authors say, "In order to cover the entire search space, they often train and evaluate a large number of architectures, leading to tremendous computation cost." This is not true. Gradient-based approaches and SPOS families do not have this limitation. Please revise this description.

(Positive) This paper studies the architecture-performance prediction, which is a crucial step in NAS. As we know, inaccurate architecture-performance prediction is the cause of ineffective NAS in almost all existing NAS methods. Given that most current NAS methods are based on weight-sharing NAS that also use shared weights to perform architecture-performance prediction, and given that many NAS methods are not better than the random architecture selection (suggested by two ICLR 2020 papers and many ICLR 2021 submissions), analyzing the architecture-performance prediction is an important step to rescue the NAS community. This paper takes a deep insight into the architecture-performance prediction, which provides a timely metric for evaluating NAS's effectiveness.

Overall, this paper provides a timely analysis of the current NAS's ineffectiveness.  I recommend accepting this paper to rescue the NAS community.

I believe this paper deserves acceptance. As we know, variants of efforts have been made to improve NAS's effectiveness since 2016, and a great process has been reached. Despite the high expectation and solemn devotion, NAS's effectiveness is believed to be still low. This is inconsistent with many pioneer researchers' expectations four years ago, in which NAS is expected to be another revolutionary technique similar to 2012's deep learning. Currently, there are many NAS papers published every year. But their effectivenesses are unclear. Differently, this paper comprehensively analyzes the architecture-performance prediction, which provides a timely analysis of the current NAS's ineffectiveness. I think this paper can attract the community's attention, encouraging the community to pay attention to the architecture-performance prediction in NAS, especially when reviewing a NAS paper. Therefore, I recommend an acceptance for this paper.

(Positive) The claimed improvement over its ICLR/ICML submission version is clear and significant. The improved version deserves acceptance by NeurIPS. Specifically, the authors have added discussion on the relationship with Bayesian Optimization. The paper also adds a comparison with Semi-NAS and ImageNet results in MobileNet Search Space. The improvement is indeed significant.

(Positive) This paper should also inspire Gradient-based approaches and SPOS families because weight-sharing NAS can be regarded as using shared weights to perform architecture-performance prediction. I think this paper is a timely solution to the NAS community.

(Positive) This paper achieves state-of-the-art results on several benchmarks, especially Open Domain Search on ImageNet.

(Positive) Sufficient experiments have been conducted to prove the effectiveness of the proposed method.

(Positive) Proper principled analyses have been provided to show the validness of the proposed method.



**Time Spent Reviewing:**

7

---

> ### Author Response · Authors · 2021-08-10
> **Respond to Reviewer onF1**
>
> Q1: All Positive Points
> - Thanks for the reviewer appreciation of our paper!
>
> Q2: Incorrect claims in the motivation section
> - Thank you for pointing this out! We will fix inaccurate claims in the motivation section and add a section to discuss the pros and limitations of Gradient-based approaches and SPOS more precisely.

---

> > ### Comment · Reviewer_onF1 · 2021-08-24
> > **Thank you, authors!**
> >
> >
> > Thank you for addressing my minor concerns. I am happy to hear from you that you will fix inaccurate claims in the motivation section and add a section to discuss the pros and limitations of Gradient-based approaches and SPOS more precisely. This is a good job!

---

> > ### Comment · Reviewer_onF1 · 2021-08-24
> > **To the authors**
> >
> >
> > Also, I have read all reviewers' comments and the authors' responses. I would like to note the author that the suggestions from the other reviewers will undoubtedly benefit the paper's improvement and make the paper attract a wide range of audiences.

---

> > ### Comment · Reviewer_onF1 · 2021-09-01
> > **A timely solution to NAS**
> >
> >
> > I really thank the reviewers and ACs for the valuable and treasurable comments, which are beneficial for improving the NAS community. I also thank the authors' great effort in providing the response. I agree with Area Chair #pMGY that Bayesian optimization methods can outperform random search for NAS. But, with plenty of experiments and rich practical experience in the industry, I would like to emphasize my worry about the low efficiency of these "elegantly formulated" NAS methods. I agree that, with the help of modern NAS benchmarks, quantifiable progress could be investigated. But to some extent, these benchmarks might be far from the real-world application (i.e., open-domain search). Whether the analysis on these benchmarks could be "well" generalized to the real-world application might be an open problem. According to my experiments, their search efficiency on ImageNet is not very high. (So far, no full-image-size full-data benchmark has been built upon ImageNet. Actually, it takes a lot of time for my group to train 500 architectures on ImageNet. I would be very grateful if someone could build such a dataset for the NAS community.)
> >
> > Since there is a long way for NAS to go and the method proposed by the authors is simple, elegant, and practical, I tend to fight for acceptance for this paper. In my opinion, I prefer a practically effective NAS method rather than an "elegantly formulated" but maybe not so helpful method in practice.
> >
> >
> > I really thank Reviewer #93Rt for his/her in-depth comments. I agree with Reviewer #93Rt that using a set of weak predictors is not new. But the reviewer also acknowledges that the proposed approach is exactly a simplified version of BO, and unlike BO, the predictor doesn’t output uncertainty, and thus the authors use a heuristic to trade-off exploitation and exploration rather than using more principled acquisition functions. I personally don't think this is a shortcoming. On the contrary, I think it is an advantage. Doesn't a simple and effective method deserve acceptance? I really like ideas that are simple but work. I thank the reviewer for requiring the theoretical analyses about why the simple greedy selection approach outperforms more principled acquisition functions. Having a theoretical result is perfect. But lacking such an analysis is also OK.
> >
> > I really thank Area Chair #pMGY for referring to NAS-BOWL. I agree that it would be better to cite and discuss the connection to this paper. It would be better if the authors could provide a comparison with it. The AC said, comparing the results of the current paper (e.g., Figure 4) to those in Figure 5 of https://openreview.net/pdf?id=j9Rv7qdXjd, NAS-BOWL appears to be much better than the current paper's method, even with less than 100 samples. I am unsure whether this is true or false. In Figure 5 of the open-review paper of NAS-BOWL, the authors used NASBOWLm and NASBOWLr to denote NAS-BOWL with architectures generated from mutating good observed candidates and from random sampling, respectively. If considering the random sampling, NAS-BOWL is significantly lower than the proposed method in this paper. Maybe I am wrong, and you can point it out.
> >
> >
> > I really thank Reviewer #nK9v for requesting several critical clarifications. Especially, the comments on the NASNet and ImageNet search space are beneficial for improving the paper. I thank the authors' efforts for addressing these concerns.
> >
> >
> > I really thank Reviewer #YANc for the constructive comments. Reviewer #YANc and I share some opinions on the strength of this paper. Specifically, we both think the idea proposed in this paper is very simple and effective. I believe this is an important reason to accept this paper.
> >
> >
> > I truly thank Reviewer #eYgg for his detailed and valuable comments. I agree that this paper is a fascinating insight. I think that the novelty of this paper is sufficient.
> >
> >
> > Considering the strengths and weaknesses of the paper, I will give a rating of acceptance. Undoubtedly, the suggestions from the other reviewers will benefit the paper's improvement and make the paper attract a wide range of audiences. If this paper is unfortunately rejected, it would be good if the Area Chair could present my praise of this paper in the meta-reviews.
> >
> > Yours sincerely,

---

### Official Review · Reviewer_nK9v · 2021-07-18

**Rating:** 6
**Confidence:** 3

**Summary:**

The authors propose a progressively sampling approach to sample architecture-performance pairs for training performance predictors. The sampling subspace shrinks guided by the previous weak predictor. Experiments are conducted on NAS-Bench-101, NAS-Bench-201, and Open Domain Search (i.e. NASNet Search Space and MobileNet Search Space)

**Ethical Concerns:**

No.

**Limitations And Societal Impact:**

No.

**Main Review:**

Concerns:

1. The proposed method is not well-justified. The progressively sampling approach seems to be biased to sample good-performing architectures. It is not clear why this could yield better results since this biased sampling could cause poorly predicted performance on under-sampled regions. Does this imply a biased sampled architecture set is good for training predictors? An ablation needs to be carried out to verify this. A simple ablated experiment could be training a strong predictor on a biased sampled architecture based on oracle and comparing it with uniform sampling.

2. The authors spend quite a few sentences to motive exploitation-exploration trade-off by sampling M architectures from TopN. However, in Table 2, sampling with M=10 and TopN=100 is only 0.01 better than sampling with M=10 and TopN=10. When searching in an open domain, it incurs $\frac{TopN-M}{M}$ times computation. it is not clear how much performance gain can bring by using this exploitation-exploration strategy. And this is not evaluated in the open domain search experiments.

3. In Table 3, the proposed method is about 0.1 better than LaNAS and BONAS. For 200 queries, the standard derivation is 0.14. It is not significant enough to conclude the method is truly better.

--------- post-rebuttal 1 ---------

I raised my score from 4 to 5. I am willing to further raise my score if the Q2 is well-addressed during the rolling discussion.

--------- post-rebuttal 2 ---------

After reading the response to Q2, I decided to my score from 5 to 6. However, reproducibility is still my main concern since the code is not provided. If the paper is accepted, I hope all the details and reproducible GitHub repo should be released publicly.

**Time Spent Reviewing:**

5 hours

---

> ### Author Response · Authors · 2021-08-10
> **Respond to Reviewer nK9v**
>
> Q1: Comparison between bias sampling and uniform sampling
> - Thanks for your question. The keypoint is exactly “a good predictor does not have to be good uniformly at the whole space”, and to effectively leverage the capacity of the predictor, we only need fine-granularity modeling for the regions we’re really interested in: where good architectures reside. As implied by the specific goal of NAS, what we need from the predictor is to “jump out” and “navigate” from bad to good architecture regions (since we hardly care about the accurate ranking of apparently unpromising architectures), and then to accurately rank and dig from only the best architectures (our real target).
>
> - So, as you correctly sensed, biased sampling will likely hamper the prediction performance on the under-sampled regions. However, in the highly-structured NAS search space, those under-sampled regions are usually bad-performing regions and do not directly affect the final performance, thus not the focus of modeling either. Our goal of WeakNAS is to pursue a “strategically biased” sampling approach to pay more attention to good-performing regions, thus improving the overall sample efficiency. Please kindly check our detailed explanations in lines 46-71 and Figure 1. This main insight of our paper also seems to be well received by the other three reviewers, e.g. “I am really convinced by the motivation of this paper” “The paper/idea is well motivated”.
>
> - We also want to clarify that we named “weak predictor” because it only predicts local subspace of the search space, and thus it is associated with biased sampling. On the contrary, the previous “strong predictor” predicts the global search space and is associated with uniform sampling.
>
> - To prove the effectiveness of bias sampling as well as the effectiveness of weak predictors, we conducted the following experiment as you requested, and used the 4-layer MLP predictor with 1000 hidden dimensions (same described in Sec. 2.4.) and a 4-layer graph convolutional network (GCN) with 1000 hidden dimensions. The results are shown in the Table below.
>
> - | NAS Predictor |Sampling |  \#Queries  | Test Acc.(\%) | SD(\%) | Test Regret(\%) | Avg. Rank |
> |:----------|:-----------:|----------:|------------:|------------:|----------:|---------:|
> | MLP | Uniform |1000 | 93.59 | 0.41 | 0.73 | 2489 |
> | MLP | Biased   |1000 | 94.25 | 0.04 | 0.07 | 1.7 |
> | GCN | Uniform |1000 | 93.96 | 0.19 | 0.36 | 85.0 |
> | GCN | Biased  |**1000** | **94.28** | **0.06** | **0.04** | **1.4** |
>
> - Given the above explanations, we hope you are now better convinced that our method is well justified and solid.
>
> Q2: In Table 2, sampling with M=10 and TopN=100 is only 0.01 better than sampling with M=10 and TopN=10
> - N controls the scope of exploration while M controls the number of samples each iteration, only M will affect the true sample cost. In the NAS-Bench settings, we evaluate the performance of all architectures using NAS predictor and sort them according to their predicted accuracy, after that we choose TopN and sample M architecture from it. As a result, varying N will not incur any extra computation cost since the architectures are already sorted.
> - Similarly, In open domain settings, we evaluate the performance of 1000 randomly sampled architectures and sort them according to their predicted performance, then we select the TopN from 1000 as the candidate, by varying TopN there is no extra computation cost since the 1000 architectures are already sorted.
> - Due to the huge computational cost of training from scratch on ImageNet, during the rebuttal week we are unable to finish the same ablation on NASNet and ImageNet search space, we will try to update it during the rolling discussion period once as soon as we have the result.
>
> Q3: In Table 3, the proposed method is about 0.1 better than LaNAS and BONAS.
> - Note that the goal of NAS is to find the closest architectures to the optimal, and currently the ranking is a more important metric in NAS. Our WeakNAS is significantly better at predicting ranking of top architectures thanks to the fine-grained modeling focused on the best architecture regions.
>
> - Despite the close accuracy gap of 0.1%, the distance to the optimal (or test regret) improves from 0.23 by 40% to 0.14. More importantly, the average rank is 30.1 times better than LaNAS and 3.2 times better than BONAS, respectively, at 200 queries. This is a very notable and clear gain.
>
> - Furthermore, Table 4/5/6 also shows our advantage over BONAS/LaNAS is highly consistent, on NAS-Bench-201 all 3 subsets, NASNet Search Space, MobileNet Search space on ImageNet. That supplies further evidence of statistical significance.
>
> - We sincerely hope the above clarifications have now convinced you of our solid performance gain. Other three reviewers also acknowledged our results to be “nice””state-of-the-art” and “indeed significant”; they also agreed “Using average rank + test regret as evaluation measures makes for a compelling case” as well as “The results are good and don’t seem cherry picked”

---

> > ### Comment · Reviewer_nK9v · 2021-08-16
> > **Response to authors**
> >
> > Thanks for the efforts to address my questions. The authors solved some of my doubts. I am convinced that the paper is well-motivated. However, I still have some concerns:
> >
> > Please add the ablation on "Comparison between bias sampling and uniform sampling" (Q1) to the later version. I think this is an important ablation to justify your motivation.
> >
> > For Q2, I am still waiting for your results on the NASNet and ImageNet search space. That is important to verify the proposed exploitation-exploration strategy.
> >
> > For Q3, I agree that ranking is a better metric than accuracy. As the authors mentioned the code will be publicly available for 100% reproducibility, I am happy to try and verify the claim in the future.
> >
> > Therefore, I raised my score from 4 to 5. I am willing to further raise my score if the Q2 is well-addressed during the rolling discussion.

---

> > > ### Author Response · Authors · 2021-08-29
> > > **Respond to Reviewer nK9v**
> > >
> > > Q2: Ablation on exploitation-exploration trade-off on MobileNet search space over 50 runs:
> > >
> > > - For the ablation on open-domain search space, we follow a similar setting in [1][2][3] by (a) Construct the evaluation pool of 10K architectures by uniform sampling the whole search space (b) Apply WeakNAS in the evaluation pool to find the best performer based on the supernet accuracy (which show strong ranking correlation to train-from-scratch accuracy as suggested in [4]). (c) Train the best performer architecture from scratch. We only did (a)(b) in this ablation, since with (a)(b) the total runtime is already at 300 s/sample * 100 sample * 50 runs * 9 experiments $\approx$ 3750 GPU hrs.
> > >
> > > - WeakNAS uniformly samples M samples from TopN predictions at each iteration, thus we can adjust N/M ratio to balance the exploitation-exploration trade-off. In the following Table 1, we set the total number of queries at 100, fix $M$ at 10 and while adjusting $N$ from 10 (more exploitation) to 1000 (more exploration). We use the optimal in the 10K evaluation pool to measure the ranking and test regret.
> > >
> > > - We found WeakNAS is quite robust within the range where N/M = 2.5 - 10, achieving the best performance at the sweet spot of N/M = 5. However, its performance drops significantly (by rank), while doing either too much exploitation (N/M <2.5) or too much exploration (N/M >25).
> > >
> > > | Sampling Methods | M | TopN |  \#Queries  | SuperNet Test Acc.(\%) | Test Regret (\%) | Avg. Rank |
> > > |:----------|:-----------:|----------:|------------:|------------:|----------:|---------:|
> > > | Uniform | -   | -    | 100 | 79.0609 $\pm$ 0.0690 | 0.1671 | 94.58 |
> > > | Biased  | 10 | 10 | 100 | 79.1552 $\pm$ 0.0553 | 0.0728 | 20.69 |
> > > | Biased  | 10 | 25 | 100 | 79.1936 $\pm$ 0.0289 | 0.0344 | 4.68 |
> > > | **Biased**  | **10** | **50** | **100** | **79.2005 $\pm$ 0.0300** | **0.0275** | **4.05** |
> > > | Biased  | 10 | 100 | 100 | 79.1954 $\pm$ 0.0300 | 0.0326 | 4.63 |
> > > | Biased  | 10 | 250 | 100 | 79.1755 $\pm$ 0.0416 | 0.0525 | 10.58 |
> > > | Biased  | 10 | 500 | 100 | 79.1710 $\pm$ 0.0388 | 0.0570 | 10.80 |
> > > | Biased  | 10 | 1000 | 100 | 79.1480 $\pm$ 0.0459 | 0.0800 | 19.70 |
> > > | Biased  | 10 | 2500 | 100 | 79.1274 $\pm$ 0.0597 | 0.1006 | 33.64 |
> > >
> > > **Table1: Ablation on exploitation-exploration trade-off over 50 runs on MobileNet search space**
> > >
> > > [1] Neural Predictor for Neural Architecture Search, ECCV 2020
> > >
> > > [2] Accuracy Prediction with Non-neural Model for Neural Architecture Search, Arxiv 2021
> > >
> > > [3] FBNetV3: Joint Architecture-Recipe Search using Predictor Pretraining. CVPR 2021
> > >
> > > [4] Single Path One-Shot Neural Architecture Search with Uniform Sampling, ECCV 2020
> > >
> > > *Note that rank does not necessarily align with accuracy, since there are chances that multiple architectures share the same accuracy, in that case, we return the average of those ranks.

---

> > > > ### Author Response · Authors · 2021-09-01
> > > > **Followup regarding Q2**
> > > >
> > > > Dear Reviewer nK9v,
> > > >
> > > > Since the discussion period is approaching its end in 48 hours, we humbly consult whether you have a chance to look at our newly posted answer to Q2, and if that could make you willing to raise our score as you previously mentioned.
> > > >
> > > > Thank you!

---

> > > ### Author Response · Authors · 2021-08-31
> > > **Respond to Reviewer nK9v**
> > >
> > > Dear Reviewer nk9v,
> > >
> > > We genuinely thank you for your time and consideration! We hope our newly added response to your Q2 has addressed your remaining concern (“Ablation on exploitation-exploration trade-off on MobileNet search space”).
> > >
> > > As the discussion period is approaching its end, we would really appreciate it if you could kindly let us know whether you have any further questions. We will be more than happy to address them fully.
> > >
> > >
> > > Yours Sincerely,
> > >
> > > Authors

---

> ### Author Response · Authors · 2021-09-05
> **To reviewer nK9v: your opinion on our response to "Q2" is high-stake**
>
> Dear reviewer **nK9v**,
>
> We feel sorry to bother you once more. We politely request again that you please look at our response to "Q2: Ablation on exploitation-exploration trade-off on MobileNet search space over 50 runs".
>
> We posted all your requested results on Aug 28th, which we believe fully addressed your concerns. As of Sep 5th, we are still waiting for your feedback on this.
>
> Especially, your feedback now becomes particularly high-stake to us, since this paper is currently within a longer-than-usual rolling discussion. The AC has kindly granted extra discussion time even beyond Sep 2nd, and it is still active. As you explicitly stated before: "I am willing to further raise my score if the Q2 is well-addressed during the rolling discussion." If you could please share your opinion here, that will be so important to us.
>
> We look forwards to hearing from you.
>
> Authors

---

### Official Review · Reviewer_93Rt · 2021-09-01

**Rating:** 4
**Confidence:** 5

**Summary:**

This work proposes a query-based NAS approach, WeakNAS, which selects the best predicted architectures at each iteration, and then updates weak predictors iteratively. By using epsilon-greedy criteria to select new query data, WeakNAS bias the training data for the predictors to good subregion of the search space, leading to better modelling of the performance of good architectures at the expense of that of bad architectures. Empirically, the authors show that WeakNAS achieves SOTA performance on a variety of search spaces. However, the novelty of the work is rather weak. As the proposed framework is behind almost every query-based NAS and can be see as a simplified BO as the authors acknowledged and the design components (sampling strategy, predictor model) themselves are not novel as well.

**Ethical Concerns:**

No applicable.

**Limitations And Societal Impact:**

The limitation and social impacts are briefly discussed in the conclusion.

**Main Review:**

**Strengths**:
1.	The proposed method, WeakNAS, achieves very impressive empirical performance i.t.o query efficiency or final test accuracy on a variety of popular search spaces and datasets. The extensive experiment efforts are highly appreciated. This gives strong evidence supporting query-based NAS.
2.	The paper is well written and the method is well explained.


**Weakness**:
The main weakness of the approach is the lack of novelty.
1.	The key contribution of the paper is to propose a framework which gradually fits the high-performing sub-space in the NAS search space using a set of weak predictors rather than fitting the whole space using one strong predictor. However, this high-level idea, though not explicitly highlighted, has been adopted in almost all query-based NAS approaches where the promising architectures are predicted and selected at each iteration and used to update the predictor model for next iteration. As the authors acknowledged in Section 2.3, their approach is exactly a simplified version of BO which has been extensively used for NAS [1,2,3,4]. However, unlike BO, the predictor doesn’t output uncertainty and thus the authors use a heuristic to trade-off exploitation and exploration rather than using more principled acquisition functions.
2.	If we look at the specific components of the approach, they are not novel as well. The weak predictor used are MLP, Regression Tree or Random Forest, all of which have been used for NAS performance prediction before [2,3,7]. The sampling strategy is similar to epsilon-greedy and exactly the same as that in BRP-NAS[5]. In fact the results of the proposed WeakNAS is almost the same as BRP-NAS as shown in Table 2 in Appendix C.
3.	Given the strong empirical results of the proposed method, a potentially more novel and interesting contribution would be to find out through theorical analyses or extensive experiments the reasons why simple greedy selection approach outperforms more principled acquisition functions (if that’s true) on NAS and why deterministic MLP predictors, which is often overconfident when extrapolate, outperform more robust probabilistic predictors like GPs, deep ensemble or Bayesian neural networks. However, such rigorous analyses are missing in the paper.


**Detailed Comments**:
1.	The authors conduct some ablation studies in Section 3.2. However, a more important ablation would be to modify the proposed predictor model to get some uncertainty (by deep-ensemble or add a BLR final output layer) and then use BO acquisition functions (e.g. EI) to do the sampling. The proposed greedy sampling strategy works because the search space for NAS-Bench-201 and 101 are relatively small and as demonstrated in [6], local search even gives the SOTA performance on these benchmark search spaces. For a more realistic search space like NAS-Bench-301[7], the greedy sampling strategy which lacks a principled exploitation-exploration trade-off might not work well.
2.	Following the above comment, I’ll suggest the authors to evaluate their methods on NAS-Bench-301 and compare with more recent BO methods like BANANAS[2] and NAS-BOWL[4] or predictor-based method like BRP-NAS [5] which is almost the same as the proposed approach. I’m aware that the authors have compared to BONAS and shows better performance. However, BONAS uses a different surrogate which might be worse than the options proposed in this paper. More importantly, BONAS use weight-sharing to evaluate architectures queried which may significantly underestimate the true architecture performance. This trades off its performance for time efficiency.
3.	For results on open-domain search, the authors perform search based on *a pre-trained super-net*. Thus, the good final performance of WeakNAS on MobileNet space and NASNet space might be due to the use of a good/well-trained supernet; as shown in Table 6, OFA with evalutinary algorithm can give near top performance already. More importantly, if a super-net has been well-trained and is good, the cost of finding the good subnetwork from it is rather low as each query via weight-sharing is super cheap. Thus, the cost gain in query efficiency by WeakNAS on these open-domain experiments is rather insignificant. The query efficiency improvement is likely due to the use of a predictor to guide the subnetwork selection in contrast to the naïve model-free selection methods like evolutionary algorithm or random search. A more convincing result would be to perform the proposed method on DARTS space (I acknowledge that doing it on ImageNet would be too expensive) without using the supernet (i.e. evaluate the sampled architectures from scratch) and compare its performance with BANANAS[2] or NAS-BOWL[4].
4.	If the advantage of the proposed method is query-efficiency, I’d love to see Table 2, 3 (at least the BO baselines) in plots like Fig. 4 and 5, which help better visualise the faster convergence of the proposed method.
5.	Some intuitions are provided in the paper on what I commented in Point 3 in Weakness above. However, more thorough experiments or theoretical justifications are needed to convince potential users to use the proposed heuristic (a simplified version of BO) rather than the original BO for NAS.
6.	I might misunderstand something here but the results in Table 3 seem to contradicts with the results in Table 4. As in Table 4, WeakNAS takes 195 queries on average to find the best architecture on NAS-Bench-101 but in Table 3, WeakNAS cannot reach the best architecture after even 2000 queries.
7.	The results in Table 2 which show linear-/exponential-decay sampling clearly underperforms uniform sampling confuse me a bit. If the predictor is accurate on the good subregion, as argued by the authors, increasing the sampling probability for top-performing predicted architectures should lead to better performance than uniform sampling, especially when the performance of architectures in the good subregion are rather close.
8.	In Table 1, what does the number of predictors mean? To me, they are simply the number of search iterations. Do the authors reuse the weak predictors from previous iterations in later iterations like an ensemble?


I understand that given the time constraint, the authors are unlikely to respond to my comments. Hope those comments can help the authors for future improvement of the paper.


**References**:
[1] Kandasamy, Kirthevasan, et al. "Neural architecture search with Bayesian optimisation and optimal transport." NeurIPS. 2018.
[2] White, Colin, et al. "BANANAS: Bayesian Optimization with Neural Architectures for Neural Architecture Search." AAAI. 2021.
[3] Shi, Han, et al. "Bridging the Gap between Sample-based and One-shot Neural Architecture Search with BONAS." NeurIPS. 2020.
[4] Ru, Binxin, et al. "Interpretable Neural Architecture Search via Bayesian Optimisation with Weisfeiler-Lehman Kernels." ICLR. 2020.
[5] Dudziak, Lukasz, et al. "BRP-NAS: Prediction-based NAS using GCNs." NeurIPS. 2020.
[6] White, Colin, et al. "Local search is state of the art for nas benchmarks." arXiv. 2020.
[7] Siems, Julien, et al. "NAS-Bench-301 and the case for surrogate benchmarks for neural architecture search." arXiv. 2020.

**Time Spent Reviewing:**

6 hours

---

> ### Author Response · Authors · 2021-09-01
> **Response to new review (which came as surprise)**
>
> Dear Reviewer 93Rt, and AC,
>
> We are truly surprised by a new review coming in **less than 48 hours before the end of the discussion**.
>
> We understand that AC has the right to recruit new reviewers, and that "If new reviews are added later on (including ethics reviews), you will have an opportunity to respond to those as well" as stated in the NeurIPS rebuttal rule.
>
> However, a review only available this late makes it **practically impossible** for any author to prepare a rebuttal in its best possible shape (unless we can be given an extension beyond 9/2 deadline to respond).
>
> We are really not sure whether this practice is fair or not in NeurIPS. We, however, will put together and post **our rebuttal (or the majority of it) in the next 24 hours** from right now.
>
> The author team already met, and determined that we are able to fully address all your concerns, if time will permit us. We will supply not only verbal arguments, but also new visual plots and result numbers when they are necessary.
>
> We politely request:
> - **Reviewer 93Rt**: please meanwhile kindly take a look at other reviews, which happened in a 3-week-long active rolling discussion and seem to have already clarified several of your questions. The overall sentiment by other regular reviewers is now quite positive about your work, as you can see
> - **AC**: very humbly, we request you to consider finalizing your recommendation, ONLY after allowing us to post rebuttals (to its best possible extent) to this last-minute review. If this new review will be taken into account for your decision, then by NeurIPS rule, we have the legitimate opportunity to respond before the deadline and be taken into your account, too.

---

> > ### Comment · Area_Chair_pMGY · 2021-09-01
> > **You will be able to respond**
> >
> > It is standard practice and very much encouraged to obtain additional reviews in borderline cases. Papers with an average score of 6 tend to have a roughly 50% acceptance rate, so this is clearly a borderline case.
> >
> > Of course, you will have the opportunity to respond. Even 72h from now is perfectly fine.
> >
> > Best,
> > AC

---

> > > ### Author Response · Authors · 2021-09-01
> > > **Re: You will be able to respond**
> > >
> > > Dear AC,
> > >
> > > Thanks very much for this swift reply and for granting us up to 72 hours. We are working on this rebuttal now and will post as soon as we ever can, since we also wish that Reviewer **93Rt** can actively participate in the follow-up discussion and raise any remaining concern before he/she could re-assess our work, if he/she might have any.
> > >
> > > We will be in touch again soon.

---

> > > > ### Comment · Area_Chair_pMGY · 2021-09-01
> > > > **Can you please post an anonymous link to a response text you can update?**
> > > >
> > > > Dear authors,
> > > >
> > > > I am actually not sure whether OpenReview will allow posts by you past September 2nd. Therefore, just in case it doesn't, could you please post an anonymous link to a text file that you can update, in order to be able to use the 72h I promised?
> > > >
> > > > Best,
> > > > AC

---

> > > > > ### Comment · Program_Chairs · 2021-09-03
> > > > > **commenting will remain enabled as needed**
> > > > >
> > > > > Commenting on all papers will remain enabled as needed.
> > > > >
> > > > > Best,
> > > > > NeurIPS 2021 Program Chairs

---

> > > > > > ### Author Response · Authors · 2021-09-04
> > > > > > **Thank you. Here we continue updating.**
> > > > > >
> > > > > > Thanks so much for the program chair to coordinate here! We also hope the **Reviewer 93Rt** has kindly checked our previous response. Below we update the comparison result with NAS-BOWLr on NAS-Bench-101, which the reviewer (and perhaps others) seem to be interested in.
> > > > > >
> > > > > > **The main message is short but clear**: under fair settings, both WeakNAS and NAS-BOWLr deliver competitive performance, but WeakNAS scales much better as an uncertainty-free, purposely simplified BO method (**showing again our motivation and strength**), in both accuracy and efficiency.
> > > > > >
> > > > > > At this moment, we think the majority of your raised questions (those that can be completed in short time) have been addressed, and we are eagerly waiting for your additional feedback or follow-up if any. We are fully prepared for the continuing discussion engagement.
> > > > > >
> > > > > > Below are our added results in detail. The original NAS-BOWLr paper only reported <= 150 queries. To match the comparison query range in our previously attached figure (https://tinyurl.com/6wx9c8kj, up to 1000 queries), we used the official codes in: https://github.com/xingchenwan/nasbowl and followed all default settings, to continue to add query numbers beyond 150.
> > > > > > - For query numbers below 150, we adopted the NAS-BOWLr authors' reported results: 93.931@100 Queries, 93.954@125 Queries, and 94.082@150 Queries. Meanwhile, WeakNAS results are: 93.429@100 Queries, 93.950 @125 Queries, and 94.110 @150 Queries. The two have comparable results.
> > > > > > - For query numbers above 150, however, WeakNAS has a notably faster increasing trend than NAS-BOWLr, with regard to the number of queries. For example, compared to its ow 150 query result as the baseline, WeakNAS increases by + 0.081 @200 queries, and +0.130@250 queries. In comparison, over its own 150 query result baseline (lower than WeakNAS already), NAS-BOWLr increases by + 0.013 @200 queries, and, +0.034@250 queries. The growing trend is clearly to the advantage of our WeakNAS.
> > > > > >
> > > > > > Therefore, we all can see now that "NAS-BOWLr performs much better" is not a true statement. Let us further comment that, as we continue to run NAS-BOWLr towards larger query numbers, we find its practical running time (official code, no change made) to grow very quickly, beyond quadratically/close to exponentially in practice. Our actual measured time costs (in the same environment from end to end) are listed FYI:
> > > > > >
> > > > > > | Queries | NAS-BOWLr RunTime (min) | WeakNAS RunTime (min) |  Speedup |
> > > > > > |:----------|:-----------:|:----------:|----------:|
> > > > > > | 25 queries | 0.5 | 0.04 | 12.5x |
> > > > > > | 50 queries|  1.4 | 0.09 | 15.5x |
> > > > > > | 100 queries | 12 | 0.19 | 63.6x |
> > > > > > | 200 queries | 96 | 0.39 | 146.1x |
> > > > > > | 300 queries | 854 (est., still running) | 0.61 | 1400x |
> > > > > >
> > > > > > ... That is why we only obtained up to 250 queries for NAS-BOWLr so far, and to complete the remaining experiments up to 1000 queries will take an unrealistically long time for our rebuttal time window. Meanwhile, WeakNAS is way more efficient and scalable by orders of magnitude.
> > > > > >
> > > > > > We thank you for bringing up this new comparison - **it actually further testifies why one would want to simplify BO and avoid full uncertainty modeling** since that appears to be hard to scale in both accuracy and efficiency. Our results provide more supports for our last Q2 claims: "WeakNAS intentionally avoided BO acquisition. It is simple, but it works better. We explain why" (please see the last post for details). The new results also nicely echo **Reviewer onF1**'s insightful point, quoted as ”I would like to emphasize my worry about the low efficiency of these elegantly formulated NAS methods.“ We will include all those new results and discussions in our final version.
> > > > > >
> > > > > > To conclude, this is a piece of work that we are very proud of, confident in, and serious about. We sincerely hope our very hard efforts to respond to this last-minute review could be seen by  **Reviewer 93Rt, AC, SAC, and PC chairs**. We wish our response be taken into everyone's accounts, wish to address any remaining question (again, we politely urge you to please ask, if any quesion), and wish a **fair** and informed decision to be made on our paper.

---

> > ### Comment · Reviewer_93Rt · 2021-09-02
> > **Thanks for being willing to respond to my last-minute review**
> >
> > Dear authors,
> >
> > I deeply appreciate your willingness to respond to my comments and I'm very looking forwards to see them. To make it easier, I'll summarise the two points that I'd like to see in the response:
> >
> > *1. Find out why WeakNAS is better than BO?*
> >
> > It can be theoretical analyses or empirical experiments. For empirical experiments, I'll suggest to use the deep ensemble version of the MLP weak predictor proposed in the paper and use the acquisition function EI. The acquisition function can be optimised by the mutation-based approach. Please follow BANANAS[1] for the set up and details. Results on NAS-Bench-301 and maybe NAS-Bench-201 would be sufficient and please present them in the plot format with mean and std instead of table format.
> >
> > *2. Compare similar base-lines in a plot format*
> >
> > Please provide the comparison between WeakNAS and the following baselines (in a descending order of priority given the time constraint) on NAS-Bench-301 and NAS-Bench-201 or 101: BRP-NAS[2], BANANAS with mutation[1], BONAS[3] (on NAS-Bench datasets, you can skip the weight-sharing evaluation) and NAS-BOWL[4]. Again, could you please provide the results in plot format with mean and std over number of queries.
> >
> > [1] White, Colin, et al. "BANANAS: Bayesian Optimization with Neural Architectures for Neural Architecture Search." AAAI. 2021.
> > [2] Dudziak, Lukasz, et al. "BRP-NAS: Prediction-based NAS using GCNs." NeurIPS. 2020.
> > [3] Shi, Han, et al. "Bridging the Gap between Sample-based and One-shot Neural Architecture Search with BONAS." NeurIPS. 2020.
> > [4] Ru, Binxin, et al. "Interpretable Neural Architecture Search via Bayesian Optimisation with Weisfeiler-Lehman Kernels." ICLR. 2020.

---

> > > ### Author Response · Authors · 2021-09-02
> > > **Response to new review [Part 2]**
> > >
> > > **Q4: Compared with NAS-Bowl and NAS-Bench-301**
> > > - We first apologize that NAS-Bench-301 experiments appear to be tight to complete given this last minute time window. We are however still actively working on it, and once we have the results we promise to update to either the anonymous link or the final camera-ready version.
> > > - Regarding comparing to NAS-Bowl, we first thank reviewer onF1 for pointing out: “In Figure 5 of the open-review paper of NAS-BOWL, the authors used NASBOWLm and NASBOWLr to denote NAS-BOWL with architectures generated from mutating good observed candidates and from random sampling, respectively. If considering the random sampling, NAS-BOWL is significantly lower than the proposed method in this paper.” We are also running NAS-BOWLr codes, to obtain more data points that can be compared with WeakNAS in a fair way. If more results become available later, they will be updated in the anonymous doc immediately.
> > > We also have to advocate that, compared to peer NAS papers in top-tier conferences, it should be fair to say our experiments are already VERY extensive: in both settings (NAS-Bench-101/201 and NASNet/MobileNet search space on ImageNet), and comparison methods (we compare with all peer works such as BONAS/BRP-NAS/LaNAS/Semi-NAS/MCTS).  This point has been unanimously recognized by other reviewers, such as “Sufficient experiments” (onF1) and as “The evaluation on the NasBench101 and 201 search spaces is thorough” (YANc).
> > > - We also have maintained a rigorous bar to shoot SOTA. For example, note that LaNAS results we compared were from their latest PAMI version updated in 2021, that were MUCH stronger than their 2019 initial arxiv version and were considered as the current best results on MobileNet space ImageNet (much more challenging for NAS Benchs). We sincerely hope the reviewer will be fair in assessing our experiment quality by taking those into account too.
> > >
> > > **Q5: Perform search on NASNet search space on ImageNet without using SuperNet?**
> > > - We agree the performance on ImageNet would hinge on both SuperNet and the search algorithm. We include the result of OFA as an apple-to-apple comparison, OFA applies evolutionary search methods and uses exactly the same supernet weight as ours. Our result in Table 6 shows our WeakNAS improved OFA from 80.0 to 81.3 with even less FLOPs, which is significantly given the already saturated performance on ImageNet.
> > > - Also, As also stated by Reviewer onF1, training sample-based methods NAS on DARTS space without using the supernet is extremely costly, for example, training only 100 models on ImageNet (each take ~500 GPU hours) would already take 50000 GPU hours as shown by previous NAS papers [5][6], thus this practice is not longer followed much in NAS literature since 2018. All SOTA works [3][4][5][6][7] now on sample-based NAS use SuperNet to evaluate performance on ImageNet, insteading of training them from scratch. We follow this convention and believe it does not discount any of our result credibility.
> > >
> > > [1] Neural architecture search with reinforcement learning, ICLR 2017
> > >
> > > [2] Learning Transferable Architectures for Scalable Image Recognition, CVPR 2018
> > >
> > > [3] ProxylessNAS: Direct Neural Architecture Search on Target Task and Hardware, ICLR 2019
> > >
> > > [4] Neural Predictor for Neural Architecture Search, ECCV 2020
> > >
> > > [5] Once-for-All: Train One Network and Specialize it for Efficient Deployment, ICLR 2020
> > >
> > > [6] Bignas: Scaling up neural architecture search with big single-stage models, ECCV 2020
> > >
> > > [7] FBNetV3: Joint Architecture-Recipe Search using Predictor Pretraining, CVPR 2021
> > >
> > >
> > > **Q6: Table 2, 3 (at least the BO baselines) in plots like Fig. 4 and 5: now provided**
> > > - We have already compared with BONAS, BRP-NAS in the paper, we’ve include the figure comparison with other BO-based methods, e.g. BANANAS, NASBOT, LaNAS, in this link (https://tinyurl.com/6wx9c8kj). It is clear now to see  the faster convergence of the proposed method.
> > >
> > >
> > > **Q7:  All mentioned “confusions”: Table 1; Table 2; and Table 3 versus Table 4 (there is no contradiction)**
> > >
> > > - Result in Table 1: what does the number of predictors mean?
> > >   - Sorry for the confusion., Yes they are equivalent to the number of search iterations. We do not reuse the weak predictors from previous iterations in later iterations, thus nothing like ensemble.
> > > - Result in Table 2
> > >   - As shown in the visualization here (https://tinyurl.com/he7zp8be), even in the good local sub-regions of NAS-Bench search space, there are still fluctuations of performance (as shown by green dot in yellow sub-region in (b)), thus certain degree of exploration is still needed to jump out of bad “caveats” and we cannot fall back on a completely “greedy” approach. As a result, uniform sampling serves as a better exploration strategy than linear-/exponential-decay (more greedy) approaches.
> > > - Results in In Table 3 contradict with in Table 4 do not contradict
> > >   - In Table 4, “samples to Global Optimal” measure the average sample to reach the optimal. In this case the search process will stop once optimal is found
> > >   - In Table3, we measure the average accuracy when limiting a certain amount of queries. In this case, the search will not stop even when the optimal is found.
> > >   - For example, say we have a search space of global optimal is 100, the first runs of queries is [25, 50, 100, 100, 100], second run of queries is [25, 100, 50, 50, 50], If we take the average of those two runs, you will find the average “sample to optimal” is 2.5 queries, while average accuracy at 5 queries is only 75, this is due to the reason that performance of predicted best architecture can possibly drop while searching.

---

> > > > ### Comment · Reviewer_93Rt · 2021-09-04
> > > > **Thanks for the detailed responses but my key concerns in my response summary haven't been addressed**
> > > >
> > > > Thanks the authors for the great efforts in producing these additional results. Deeply appreciated. However, the main questions, as summarised in my latest comment, have not been fully addressed yet. So I'll highlight them again below:
> > > >
> > > >
> > > > 1. **Regarding the core contribution and a plot comparison with BRP-NAS**
> > > >
> > > > Thanks for the clarifying that the “core contribution is re-formulation of predictor-based NAS, and then the weak predictor solution inspired by the insight of specially structured NAS space”. However, using sampling strategies to zoom into high-performing regions and then update the predictors iteratively to fit the high-performing region better (as shown in the nice visualisation on the search processes of WeakNAS) are the key idea and thinking behind almost every query-based NAS method using predictors and most query-based NAS are aware that the architecture search space is non-uniform and heterogeneous as many excellent NAS-Bench works have visually shown that.  So I’m sorry that I still don’t see the novelty of the reformulation mentioned by the authors compared to what is already been done. Even if that is not the case,  this core contribution and the methodology of this work is exactly the same as that of BRP-NAS though BRP-NAS doesn’t explicitly highlight it in their paper. In fact, as acknowledged by the authors in the response, the *only difference* between the proposed method and BRP-NAS is the sampling strategy: WeakNAS choose from the top-K only, thus more greedy but BRP-NAS would add a portion of random architectures to encourage more exploration. However, both are “highly similar to $\epsilon$-greedy” as the authors mentioned in the paper. Thus, this difference seems a very incremental contribution to me unless the author can show that this small change can lead to *significantly better* results.  I.e. as I emphasised in my summary above: **a plot comparison with BRP-NAS** (or even with WeakNAS using BRP-NAS sampling strategy) would be helpful to see the gain.
> > > >
> > > >
> > > >
> > > > 2. **Compare to the BO variant of WeakNAS**
> > > >
> > > > Regarding the BO variant of WeakNAS, I appreciate that the authors are“ trying to implement something like that now”. As I highlighted in my response summary above, this is not trying to ask for another baseline (which the authors have already done a great job) but *a necessary and important ablation* to show convincingly that the proposed simplification (no uncertainty in modelling and heuristic sampling strategy) is indeed better than its BO variant when compared apple-to-apple (i.e. use the same weak predictor as WeakNAS and get uncertainty using deep ensemble or BLR, then applied it with EI). Just a quick note on the uncertainty modelling, it is much simpler than it sounds and adds little complication to the predictor. For example, after training the MLP, you just need to replace the final layer of it with BLR to get uncertainty. The fitting of BLR is almost instant and all the hyperparameters are self-tuned by maximising the marginal likelihood. As for deep ensemble, you just need to train the same predictor 3 times with different initialisations. When conducted in parallel, this doesn’t increase training time at all and when conducted in sequence, the increased overhead is still negligible compared to the true bottleneck in NAS which is the evaluation of the architectures. As for the acquisition function like EI, it’s a analytic expression which can be coded up in one line and doesn’t need to tune any $\alpha$ parameter. Therefore, to me, simplifying such already simple and neat methods (simple uncertainty modelling and simple acquisition function) doesn’t not make a significant contribution unless it leads to significant performance gain. So the proposed ablation would be a valuable evidence to strengthen your claimed contribution. Of course, a theoretical justification would be a good alternative.
> > > >
> > > >
> > > > 3. **Other minor comments**
> > > >
> > > > Thanks a lot for comparing to NAS-BOWLr (random sampling). I was curious why the authors don’t compare to NAS-BOWLm (mutation), which outperform NAS-BOWLr by a large margin and uses a more greedy sampling strategy, thus closer to the setting of WeakNAS. However, I totally agree that GP is less scalable to NN when the number of queries is large, and comparison to NAS-BOWL is less important (lower priority) than the comparison to BO-version of WeakNAS and BRP-NAS as I explicitly highlighted in my response summary. Apologies for my misunderstanding on BO-NAS and thanks for correcting that.

---

> > > > > ### Author Response · Authors · 2021-09-05
> > > > > **We are working on new response**
> > > > >
> > > > > Dear reviewer **93Rt** (and AC):
> > > > >
> > > > > We got your questions. We have now been focused on your questions 1 and 2 in the last post (BRP-NAS, BO variant of WeakNAS) -  both we believe we are able to fully address with experiments.
> > > > >
> > > > > However, a server outage today has unfortunately delayed us (due to reasons beyond our control). We have to re-launch everything just now and we target posting our next response (or the majority of it) within 24 hours from now.
> > > > >
> > > > > Please stay tuned if that is still possible. We will be back in touch as soon as we can.

---

> > > > > ### Author Response · Authors · 2021-09-05
> > > > > **Response to new review**
> > > > >
> > > > > **Q1: Comparison to BRP-NAS: FINISHED**
> > > > >  - We use the BRP-NAS official implementation [1] and use the config: configs/predictors/acc_gcn_nasbench101_bin.yaml on NAS-Bench-101. We **did not alter any hyperparameter** except the total number of queries. We average over 6 runs and calculate the mean and standard deviation. As you required, we show the plot in this link (https://tinyurl.com/ct4y3kkw). You can see WeakNAS’s performance is clearly superior to BRP-NAS, even using much simpler predictors such as MLP or Gradient Boosting Tree.
> > > > >
> > > > >  - We also find that BRP-NAS runtime is significantly slower than ours (possibly due to the GCN predictor). We list the runtime comparison in the Table below
> > > > >
> > > > >  - | Queries | BRP-NAS RunTime (min) | WeakNAS RunTime (min) |  Speedup |
> > > > > |:----------|:-----------:|:----------:|----------:|
> > > > > | 20 queries   | 3.13 | 0.04 | 78.25x |
> > > > > | 50 queries   | 23.9 | 0.09 | 265.5x |
> > > > > | 100 queries | 77 | 0.19 | 405.2x |
> > > > > | 200 queries | 516 | 0.39 | 1323.1x |
> > > > >
> > > > > [1] https://github.com/SamsungLabs/eagle
> > > > >
> > > > > **Q2: Compare to the BO variant of WeakNAS: Updating**
> > > > >  - We use the following setting in our WeakNAS BO variant on NAS-Bench-101
> > > > >    - a) We obtain the variance (uncertainty) measure using deep ensemble of 5 MLPs (same MLP used in WeakNAS)
> > > > >    - b) We then apply Expected Improvement (EI) acquisition function given the mean and variance (uncertainty) calculated in (a) using this implementation [1].
> > > > >  - We average over 4 runs and calculate the mean and standard deviation. However, this experiment is still ongoing due to the fact that computing EI acquisition function is very costly (needing to calculate CDF and PDF of each sample over all 423K samples). We aim to update more results in a few more hours.
> > > > >
> > > > > [1] https://github.com/naszilla/naszilla/blob/5575cc8c95e79ce5743e8ea7ef53d6da900f8480/naszilla/acquisition_functions.py#L20

---

> > > > > ### Author Response · Authors · 2021-09-06
> > > > > **Response to new review [Adding FINISHED experiment on BO variant of NAS]**
> > > > >
> > > > > **Q2: Compare to the BO variant of WeakNAS: FINISHED**
> > > > >
> > > > >  - We use the following setting in our WeakNAS BO variant on NAS-Bench-101
> > > > >    - a) We obtain the variance (uncertainty) measure using deep ensemble of 5 MLPs (same MLP used in WeakNAS)
> > > > >    - b) We then apply Expected Improvement (EI) acquisition function given the mean and variance (uncertainty) calculated in (a) using this implementation [1]. We average over 4 runs and calculate the mean and standard deviation.
> > > > >
> > > > >  - (1) We found computing EI acquisition function is extremely costly (since it need to calculate CDF and PDF of each single sample over all 423K samples), we list the runtime comparison in the Table below.
> > > > >
> > > > >    - | Queries | WeakNAS EI variant RunTime (min) | WeakNAS RunTime (min) |  Speedup |
> > > > > |:----------|:-----------:|:----------:|----------:|
> > > > > | 20 queries   | 12.0 | 0.04 | 300x |
> > > > > | 50 queries   | 31.7 | 0.09 | 352.2x |
> > > > > | 100 queries | 65 | 0.19 | 342.1x |
> > > > > | 200 queries | 212 | 0.39 | 543.6x |
> > > > > | 500 queries | 375 | 1.15 | 326.1x |
> > > > >
> > > > >  - (2) We show the plot in this link (https://tinyurl.com/hx52pjc), you can see our uncertainty-free Top sampling acquisition function is clearly superior to uncertainty-based EI acquisition function, even with up to 300x faster in runtime.
> > > > >  - (3) Moreover, our uncertainty-free (probabilistic) acquisition function can be formally defined as the step function shown below,
> > > > >
> > > > >    - $y = 1/N, \\quad x \\geq \\theta$
> > > > >
> > > > >    - $y = 0, \\quad x < \\theta$
> > > > >
> > > > >  - where at each iteration we sample $M$ samples from Top$N$, $\theta$ is the threshold of splitting the whole set into Top$N$ and the rest.
> > > > > We found the uncertainty-free acquisition function is much simpler and efficient then EI acquisition function, and works better in practice.
> > > > >
> > > > > [1] https://github.com/naszilla/naszilla/blob/5575cc8c95e79ce5743e8ea7ef53d6da900f8480/naszilla/acquisition_functions.py#L20

---

> > > ### Author Response · Authors · 2021-09-02
> > > **Response to new review [Part 1]**
> > >
> > > We summarize the main questions of Reviewer 93Rt and organize our answers in the following seven aspects. We believe we have addressed every raised question thoroughly, and we invite you to raise any additional question for a fair rolling discussion as already approved by AC.
> > >
> > > Following the recommendation by AC, we also include an anonymous link to a document that is currently empty:
> > >
> > > https://tinyurl.com/4aznyen4
> > >
> > > If OpenReview won’t allow us to update beyond 9/2 EOD, we will keep updating the doc to reflect our response to every new question from you. Every possible effort will be made at our side, to make sure our work receives fair treatment and evaluation.
> > >
> > > We also noticed that Reviewer onF1 provided very kind and supportive comments during our preparation, which we deeply appreciate!
> > >
> > > **Q1. The Novelty of WeakNAS**
> > >
> > > - The main merit of WeakNAS lies exactly in its intentional “simplicity”, and insight behind so
> > >
> > >   - We have never claimed to be the first to  iteratively refine predictors, or to have invented any new predictor: they are NOT the point of the paper. Instead, The principled re-formulation of predictor-based NAS, and then the weak predictor solution inspired by the insight of specially structured NAS space (explained below), makes our actual core contribution. Our claim has been well-validated by our comprehensive empirical comparison and “very impressive performance” on both NAS-benches and open-domain search.
> > >   -The crucial insight of WeakNAS is, due to the special structure of NAS search space, its search strategy can be largely simplified. Specifically, a line of progressively evolving weak predictors can be found to connect a path to the best architecture.
> > >   -We have included a new visualization to help you understand: https://tinyurl.com/he7zp8be As explained in the figure captions, WeakNAS build a iterative process, where it searches for some top-performing cluster at the initial search iteration and then “zoom-in” the cluster to find the top performers within the same cluster
> > >   -The unique insight was the reason why we could leverage WeakNAS to navigate through the highly-structured search space. Hence, it requires only to estimate a series of weak predictors to fit small local spaces and to progressively move the search space towards the subspace where good architecture resides.
> > >   -For the same reason, it is in fact treated as an advantage that we demonstrate WeakNAS can work effectively and achieve SOTA with existing “old” and “simple” predictors (MLP, trees..), rather than any “new” and “stronger” predictors. The main message is: they are sufficient to work well in our new scheme. Indeed, our framework is general and can be used with various predictors or architecture encodings.
> > >   -As endorsed by Reviewer onF1 “a heuristic to trade-off exploitation and exploration is not a shortcoming, on the contrary it is an advantage. Doesn't a simple and effective method deserve acceptance?” We strongly agree with Reviewer onF1 that the NAS community indeed needs a practically effective NAS method, perhaps more  than an "elegantly formulated" but maybe not scale in practice. We will explain more regarding “why not BO” in our Q2
> > >
> > > - Our difference with BRP-NAS
> > >   - Our sampling strategy is NOT exactly the same as that in BRP-NAS. Given a sample budget of $M$, BRP-NAS picks both $N$ samples from Top-$K$ and $(M-N)$ random models from the entire search space, while our WeakNAS only picks random $M$ models in Top-$K$, thus is a more “greedy” strategy.
> > >   - BRP-NAS controls the exploitation and exploration trade-off by adjusting $\alpha$=$\frac{M-N}{M}$, and did not have any section or ablation discussing the exploitation and exploration trade-off: they only empirically chose $\alpha$=$0.5$ as the default ratio. On the other hand, our WeakNAS control the exploitation and exploration trade-off by adjusting $\frac{K}{M}$ ratio, and we did a comprehensive analysis on the exploitation and exploration trade-off on both NAS-Bench and MobileNet Search Space on ImageNet (please see the latest nK9v response to reviewer).
> > >
> > > **Q2. WeakNAS intentionally avoided BO acquisition. It is simple, but it works better. We explain why.**
> > >
> > > - Our method can be regarded as a purposely simplified variant of Bayesian Optimization (BO): the weak predictors take a similar role to typical acquisition functions, but ours refer to no explicit uncertainty-based modeling such as Gaussian Process (which are often inefficient to scale up). Section 2.3 has analyzed the connection and differences. Our current approach does not rely on any explicit uncertainty-based modeling such as Gaussian Process, mainly motivated by reducing the search cost: uncertainty modeling is heavier than deterministic predictors and harder to scale in practice. However, our approach is not free of exploration – just that we have a different, simpler build-in exploration mechanism by trading offer N versus M.
> > >
> > > - We find our simplified exploration in this current setting works well for the NAS application, presumably due to the structured NAS search space. Overall, NAS spaces in use often have a highly non-uniform distribution of architectures, whose density is also heterogeneous (although a precise definition of architecture distribution or density hinges on the concrete definition of distance). That is because the candidate architectures were created from varying operators (often causing similar performers), to varying width or depth (often causing bigger performance gaps).
> > >
> > > - Therefore, recent work [1] suggests the architectures display highly clustered distribution; and the best performers are often gathered close [1] or follow some specific patterns [2]. This also leads to our work’s underlying assumption: we can progressively connect the line from the initialization to the finest subspace, where the best architecture resides. We also show a t-SNE visualization of NAS search space here (https://tinyurl.com/he7zp8be) to better illustrate the search dynamic of our simple heuristic trade-off of exploitation and exploration.
> > >
> > > - Based on the insight of the non-uniform or heterogeneous search space, the demand for the exploitation-exploration trade-off is also heterogeneous across different search stages. At the beginning stage, more exploration will be needed to identify the promising (shown as the yellow regions) direction towards the finer space. That is naturally achieved because the weak predictor at the initial stage only roughly fits the whole space (in other words, it CANNOT fit too well by then, and we intentionally make so). As we keep zooming in to the good-performing subspace, the weak predictor is also gradually refined and better fit, therefore making stronger exploitation. Eventually, our weak NAS predictor is made to adaptively balance the exploration and exploitation throughout the search process, which echoes the findings by many prior works such as [3][4][5].
> > >
> > > - Your suggestion “to modify the proposed predictor model to get some uncertainty (by deep-ensemble or add a BLR final output layer) and then use BO acquisition functions (e.g. EI) to do the sampling” is interesting. We are trying to implement something like that now (it will take a longer time, and we will report in an anonymous doc or final version if ready). However, we also respectfully remind that, this paper intentionally AVOIDs going with BO or uncertainty modeling with significantly more complicated predictors (see all above explanations). Our main message is to show that even equipped with weak “deterministic MLP predictors, which is often overconfident when extrapolating”, our framework can still work well. Therefore, we consider your suggestion as a nice addition to have, but not a necessity.
> > >
> > > - This simplicity is also recognized by reviewer onF1 “the method proposed by the authors is simple, elegant, and practical” , and he/she also suggested “a heuristic to trade-off exploitation and exploration is not a shortcoming, on the contrary it is an advantage”. We are again thankful for the reviewer onF1.
> > >
> > > [1] Exploring the Loss Landscape in Neural Architecture Search, Colin White, Sam Nolen, Yash Savani
> > >
> > > [2] Ilija Radosavovic, Raj Prateek Kosaraju, Ross Girshick, Kaiming He, Piotr Dollár, Designing Network Design Spaces
> > >
> > > [3] "AlphaX: eXploring Neural Architectures with Deep Neural Networks and Monte Carlo Tree Search." AAAI 2019.
> > >
> > > [4] "Sample-efficient neural architecture search by learning action space." TPAMI 2021.
> > >
> > > [5] "Revisiting Neural Architecture Search." arXiv 2020.
> > >
> > > **Q3. Our comparison with BONAS is fair and convincing - and you perhaps misread BONAS details.**
> > > - “BONAS uses a different surrogate which might be worse than the options proposed” - this is not true
> > >   - BONAS used three kinds of different surrogate models, GCN, MLP, LSTM and MetaNN. Take MLP as a example, BONAS use a 4-layer MLP of hidden size (512, 2048, 2048, 512)), compared to WeakNAS that uses the same 4-layer MLP of similar hidden size (1000, 1000, 1000, 1000), we have empirically compared the two MLPs on NAS-Bench-101, and have found they yield similar performance in terms of ranking in terms of ranking architectures.  In fact, the best performing BONAS use GCN as a surrogate model, which is apparently stronger han our 4-layer MLP, However, using our simple heuristic of exploitation and exploration trade-off, WeakNAS are able to surpass BONAS even with a weaker predict model. Therefore, it is clear that our comparison never put BONAS in any unfair disadvantage.
> > > - “BONAS use weight-sharing superNet” - this is also not true.
> > >   - We politely ask the reviewer take a closer look of the BONAS paper: the experiment of BONAS on NAS-Bench-101/201 did not use any weight-sharing supernet, it directly use Ground-Truth of NAS-Bench to train the predictor (mentioned in Sec 4.2).

---

### Author Response · Authors · 2021-09-07
**Summary and Reminder: Our updates since 9/1 wait to be considered**

Dear **AC** and **Reviewer 93Rt**,

We are writing to remind your attention again, to the new responses that we have prepared with very hard efforts, in reply to *Reviewer 93Rt*'s new comments posted on Sept 1st.

Per AC/PC's guidance, we took our fair share of extra time and posted comprehensive answers to addressing all raised questions. In particular, we have recently included the key new results to comparing with BRP-NAS, and BO-variant WeakNAS. Those results were posted 27 hours ago, and we are prepared to answer any feedback or additional comment.

Very sincerely, we want to ensure that our new responses will be checked and taken in accounts by *Reviewer 93Rt* and *AC*, before a final decision is reached. We believe those are very essential in order for a fair chance and an informed decision for our paper.

Best
Authors,

---

> ### Comment · Area_Chair_pMGY · 2021-09-10
> **Can you make the code available now?**
>
> Dear authors,
>
> Thank you for the various comparisons you provided. I've discussed with Reviewer 93Rt, and we're both very surprised by the strong results you show compared to BRP-NAS and BO variants given the minor differences. We do agree that if those improvements hold up this finding would be valuable to share with the community.
>
> Since the foremost reason to accept this paper would be the strong empirical results, it it very important for the community to be able to reproduce your results. While you already promise to share code upon acceptance, some authors mean different things than others by "sharing code", e.g., some only share final architectures. Rather, I'd like to ask you to also share all details, such as your training pipelines, hyperparameters, etc, as advocated by the NAS best practice checklist (https://jmlr.org/papers/v21/20-056.html).
>
> I would like to ask you to share this code in private form for review purposes. Please do ensure to include all of the following (in order of importance):
> - Code for your NAS method, including hyperparameters and information on how the hyperparameters were tuned.
> - Code for your training pipelines (i.e., including the supernet) and final evaluation pipelines for the open domain search benchmarks.
> - Code/scripts used to run all baseline methods
> - Code/scripts for reproducing all tables and figures
> - Final architectures found by the methods
> - Anything else you can think of that a reviewer/reader would need to reproduce your results.
>
> If you can provide this I would recommend acceptance, since it would allow the NAS community to build on it and find out which components exactly are important to yield better performance than the baselines.
>
> Best,
> AC

---

> > ### Author Response · Authors · 2021-09-14
> > **Code Release: Part I**
> >
> > We've been working really hard in the past two days to clean and pack our codes. The process takes lots of time due to the very large number of experiments involved in your requests.
> >
> > Please now find the first batch of our code release: WeakNAS and variants on NAS-Bench-101 here (https://tinyurl.com/3x7hmhcm). That includes training pipelines, hyperparameters, evaluation, and figure/table reproductions. Besides, we also included the searched final architectures on open domain MobileNet ImageNet search.
> >
> > We are continuing the code cleaning process and we plan to release more batches as they become ready. Meanwhile, let us publicly promise here in an explicit way: all your requested items will become public in less than three weeks, after this paper is accepted.
> >
> > While we are doing every possible effort to answer to your requests, let us all please also acknowledge that. what you request demands a huge code repository. For an accepted paper to NeurIPS or equivalent, it is normal to take a few weeks to prepare all codes ready. We hope you consider this fact too.

---

> > > ### Author Response · Authors · 2021-09-20
> > > **Code Release: Part II**
> > >
> > > Please kindly find the second batch of our code release here (https://tinyurl.com/2722r6b3). We release (i) WeakNAS and variants on NAS-Bench-201, which includes training pipelines, hyperparameters, evaluation, and figure/table reproductions; and (ii) Besides, we also included the training and final evaluation pipelines in open domain MobileNet Search Space, which consist of three parts: (a) SuperNet training (b) Search (c) Train-from-Scratch.
> > >
> > > We are continuing the code cleaning process and we plan to release more batches as they become ready. We affirm our promise again, that ALL your requested items will become public in less than three weeks after this paper is accepted.
> > >
> > > As we are aware that the reviewer discussion + AC/SAC discussion are approaching the end, we would like to inquire if the AC and Reviewer 93Rt have checked our released codes. We prepared them with hard efforts to follow your guidance that it "would allow the NAS community to build on it and find out which components exactly are important to yield better performance than the baselines." If any concern still remains that might prohibit a positive recommendation of this work, we would appreciate if you could let us know now.

---

### Decision · Program_Chairs · 2021-09-28

**Decision:**

Accept (Poster)

**Comment:**

This paper introduces a query-based NAS method that is a simplified version of Bayesian optimization and is also very similar to BRP-NAS. Novelty is therefore low. On the upside, surprisingly, the method performs very well compared to many baselines that were requested during the rebuttal period. After the initial set of reviews, the paper had borderline scores, and the internal reviewer discussion was not moving further. I therefore recruited an emergency reviewer, who first gave a rejecting score (citing lack of novelty, and not expecting major performance improvements over the many similar previous methods). However, the authors provided strong results. If the code is made available for fully reproducing the work I am in favour of accepting the paper as this would allow studying *why* the proposed method performs better than Bayesian optimization etc, and since it would also provide a solid available baselines for future advances.
Specifically, I asked the authors to make available the following:
- Code for their NAS method, including hyperparameters and information on how the hyperparameters were tuned.
- Code for their training pipelines (i.e., including the supernet) and final evaluation pipelines for the open domain search benchmarks.
- Code/scripts used to run all baseline methods
- Code/scripts for reproducing all tables and figures
- Final architectures found by the methods
- Anything else they can think of that a reviewer/reader would need to reproduce your results.


**Consistency Experiment:**

NeurIPS has a long history of experimentation. In 2014, NeurIPS ran an experiment in which 10% of submissions were reviewed by two independent committees to quantify the randomness in the review process. This year, we repeated a variant of this experiment to see how the quality of the review process has changed over time.  This paper was part of the experiment and was therefore assigned to two committees (consisting of reviewers, an Area Chair, and a Senior Area Chair) that reached independent decisions.  If both committees made the same recommendation, this recommendation was followed. If a single committee recommended acceptance, the paper was accepted (with the exception of a few cases in which the other committee identified what we considered a fatal flaw, e.g., an error in a key result).

Both committees reached the same decision: **Accept (Poster)**

The other committee assigned to the paper recommended **Accept (Poster)**.  You can find the other set of reviews, along with any follow up discussion with the authors here:
https://openreview.net/forum?id=chwaxchpG3